# CAOTE: KV Caching through Attention Output Error based Token Eviction

## Abstract

While long context support of large language models has extended their abilities, it also incurs challenges in memory and compute which becomes crucial bottlenecks in resource-restricted devices. Token eviction, a widely adopted post-training methodology designed to alleviate the bottlenecks by evicting less important tokens from the cache, typically uses attention scores as proxy metrics for token importance. However, one major limitation of attention score as a token-wise importance metrics is that it lacks the information about contribution of tokens to the attention output. In this paper, we propose a simple eviction criterion based on the contribution of cached tokens to attention outputs. Our method, CAOTE (KV **C**aching through **A**ttention **O**utput error based **T**oken **E**viction), optimizes for error due to token eviction, by seamlessly integrating attention scores and value vectors. This is the first method to use information from the value vector on top of attention-based eviction scores. Additionally, CAOTE can act as a meta-heuristic method with flexible usage with any token eviction method. We show that CAOTE, when combined with state-of-the-art attention score-based methods, always improves accuracies on the downstream task for LLAMA3 and QWEN2.5 model families, indicating the importance of leveraging information from values during token eviction process.

## 1 Introduction

Large Language Models (LLMs) represent a large step forward in natural language processing by demonstrating remarkable proficiency in tasks such as text generation (1), machine translation (2), and question-answering (3). Many of these tasks require handling long prompt inputs efficiently, particularly in retrieval-augmented generation (RAG), long-form document understanding (4), summarization (5), and multi-turn dialogue systems (6)—collectively referred to as long-context LLMs. A key challenge in long-context applications is the increased latency during inference and generation, primarily due to the quadratic computational complexity of self-attention and the growing memory demands of handling long sequences.

To alleviate computational overhead, Key-Value (KV) caching is a widely adopted technique that enables faster inference in LLMs. It does so by storing the key-value states of previously processed tokens and reusing them when generating new tokens, reducing redundant computations (7). However, while KV caching significantly enhances efficiency, it comes at the cost of substantial memory consumption as mentioned in (8) , especially in long-context setups. In such cases, the memory footprint of the KV cache often exceeds the memory usage of the model itself, making it a major bottleneck, especially, for memory-constraint devices. In this paper we focus on improving KV cache eviction methods by keeping relevant KV to utilize the memory budget in the best possible way.

Several methods have been proposed to optimize KV cache memory usage, including sparse attention mechanisms that selectively attend to a subset of tokens rather than all previous tokens reducing memory and computational complexity (9; 10), efficient attention architectures such as linearized attention (11), and memory-efficient transformers (12) which approximate self-attention to minimize memory consumption. These methods require from-scratch training or model finetuning, however, in this work, we focus on post-training token eviction method which dynamically discard less important tokens from the KV cache to control memory growth (8). Token eviction methods offer a distinct advantage in memory management by explicitly reducing the KV cache size while preserving critical information. These methods leverage the sparsity of attention — a phenomenon where a small subset of cached keys contribute disproportionately in the attention mechanism — to selectively retain only the most important tokens.

Typically, token importance is determined based on attention scores or, equivalently, attention weights, which measure the alignment between query and key tokens (8; 13). However, since the attention output is a linear combination of attention weights and value vectors, evicting tokens simply based on attention scores may lead to suboptimal decisions, as it does not consider the contribution of value vectors from the tokens to be evicted.

In this paper, we propose *CAOTE*, a post-training cache eviction method that seamlessly integrates eviction (attention) scores with value vectors by optimizing the eviction error, unlike recent methods, as shown in Table 1. *CAOTE* offers flexibility by being applicable to any post-training eviction method and also includes an efficient counterpart. When combined with recent cache eviction methods (8; 13; 14), we observe performance boost for all recent token eviction methods on a variety of downstream tasks: 16 tasks from LongBench, Needle-in-Haystack and Perplexity measurement tasks.

The paper is divided into the following sections: Section 2 discusses background for token eviction. Our main method is in Section 3 which consists of *CAOTE*. In Section 4, we present experiments and results. Section 5 consists related works and Section 6 consists the conclusion.

## 2   Background

Token eviction is a popular methodology (10; 13; 8) for decoder-only transformer inferences that prevents KV-Cache from growing linearly as token generation continues by preventing less important tokens from being cached. This has dual benefits; first, it limits memory consumption for the KV-cache and second, it reduces the computational complexity of the attention mechanism. Here, we consider the case of processing the input prompt block-wise in resource-restricted environments. In this case, token eviction can save memory and computation not only in the generation phase, but also in the prefill phase, leading to a shorter time-to-first-token which is especially beneficial when the input prompt is extremely long.

| Method | Keys | Values | Min eviction error |
|--------|------|--------|--------------------|
| H2O | ✓ | ✗ | ✗ |
| TOVA | ✓ | ✗ | ✗ |
| SnapKV | ✓ | ✗ | ✗ |
| X+CAOTE | ✓ | ✓ | ✓ |

Table 1: Overview of recent token eviction methods compared to CAOTE based on components used during eviction.

With a sequence of hidden states $X^l = [x_1^l, \ldots x_t^l] \in \mathbb{R}^{t \times d}$, the transformer block updated the hidden states as follows:

$$X^{l+1} = \Phi_{\text{TRANS}}^l(X^l) = \phi_{\text{FF}}^l\left(\phi_{\text{SA}}^l(X^l)\right) \tag{1}$$

where, $x_j^l$ is the hidden state of token $j$, $\phi_{FF}^l$ denotes the feedforward layer, and $\phi_{SA}^l$ denotes the self-attention layer, superscript $l$ denotes the layer index.

For brevity, we omit normalization layers and skip connections.

**Prompt prefill**   Given hidden-states $X^l \in \mathbb{R}^{t \times d}$ of $t$ tokens, the self-attention layer process inputs as follows:

$$X_{\text{attn}}^l = \phi_{\text{sa}}(Q^l, K^l, V^l) = \underbrace{\text{Softmax}(Q^l(K^l)^\top)}_{A^l} V^l \tag{2}$$

where, $Q^l, K^l, V^l \in \mathbb{R}^{t \times d}$ and $A^l \in \mathbb{R}^{t \times t}$. Here, we omit output layer projection and multi-head extention for brevity.

**Block-wise prompt prefill**   Instead of processing all tokens at once (resulting in attention matrix: $A^l \in \mathbb{R}^{t \times t}$), we can process tokens in block-size $m$, which also helps in evicting tokens in small blocks instead of larger chunks.

$$X_{\mathrm{attn},t+1:t+m}^l = \underbrace{\mathrm{Softmax}(Q_{t+1:t+m}^l [K_{:t}^l, \mathbf{K_{t+1:t+m}^l}]^T)}_{A^l \in \mathbb{R}^{m \times (t+m)}}[V_{:t}^l, \mathbf{V_{t+1,(t+m)}^l}] \tag{3}$$

where, the new token hidden states are $X_{t+1:t+m}^l$ which are projected to $Q_{t+1:t+m}^l, K_{t+1:t+m}^l, V_{t+1:t+m}^l$

**Generation**. In autoregressive generation a single token is generated at each iteration

$$X_{\mathrm{attn},t+1}^l = \underbrace{\mathrm{Softmax}(Q_{t+1}^l [K_{:t}^l, \mathbf{K_{t+1}^l}]^T)}_{A^l \in \mathbb{R}^{1 \times (t+1)}}[V_{:t}^l, \mathbf{V_{t+1}^l}] \tag{4}$$

For resource-constraint hardware, single-inference KV cache prefill for a large number of input tokens may cause out-of-memory error or slow throughput. On the other hand, combining block-wise prefill with token eviction after processing each block of prompt can resolve this issue and improve throughput (15; 16). For a block-size $m$, when $b$ tokens are initially processed, the usage of memory and computation power can always be kept within budget constraints by processing the next $m$ tokens and evicting the next $m$ tokens. In this case, attention matrix has size: $A^l \in \mathbb{R}^{m \times (b+m)}$.

Recent eviction methods use variants of attention scores from $A^l$ for evicting tokens by using a function (or operator) to map $f_{\mathrm{score}}(A^l) : \mathbb{R}^{m \times (b+m)} \to \mathbb{R}^{b+m}$, where $f_{\mathrm{score},j}$ is the retention score (or score) for token $j$, the top-$b$ tokens are retained based on the score: $\underset{b}{\mathrm{argmax}}\, f_{\mathrm{score}}(A^l)$, where $b$ is the budget (maximum tokens allowed per layer). Examples of score functions, for *H2O*, $f_{\mathrm{score},j} = \Sigma_{i=1}^m A_{i,j}$, and for *TOVA*, $f_{\mathrm{score},j} = A_{-1,j}^l$. The process of token eviction follows intuitive steps as shown: computing scores for newly processed tokens, choosing top-$b$ tokens, computing attention output using the top-$b$ tokens' hidden-state, we show the steps below:

$$A_{b+m}^l = \mathrm{Softmax}(Q_{b+1:b+m}^l [K_{:b}^l, \mathbf{K_{b+1:b+m}^l}]^T) \tag{5}$$

$$i_1, \ldots, i_b = \underset{j \in \{1,\ldots,b\}}{\mathrm{argmax}}\, f_{\mathrm{score},j}(A_{b+m}^l) \tag{6}$$

$$X_{\mathrm{attn}}^l = \mathrm{Softmax}(Q_{b+1:b+m}^l (K_{i_1:i_b}^l)^T) V_{i_1:i_b}^l \tag{7}$$

where the key in bold are correspond to the new tokens' hidden states being inserted. In above equation we assume that no new query token was evicted for ease of notation. During generation, the flow remains same with $m = 1$.

## 3   CAOTE: KV Caching through Attention Output-Based Token Eviction

Our method is developed based on two key insights: (i) existing token eviction policies primarily rely on attention scores derived from queries and keys, and (ii) attention output is a linear combination of values. We find that optimizing for eviction error is same as change in attention output due to eviction, which can be computed in closed-form for each token during generation and can be used as the eviction score (*CAOTE* score).

We first introduce *CAOTE* in Subsection 3.1 and how to compute eviction error in closed-form. This is followed by a discussion of its meta-property, demonstrating its applicability with other score-based attention methods such as H2O (8) in Subsection 3.2. Finally, we propose an efficient approximation of *CAOTE* in Subsection 3.3. The general workflow of *CAOTE* is illustrated in Fig. 1, highlighting that the modifications to existing token eviction methods are minimal.

### 3.1   CAOTE Score

The objective of our token eviction is to minimize eviction error: the change in attention output before and after eviction. We formulate eviction error for the generation scenario in which a single new token

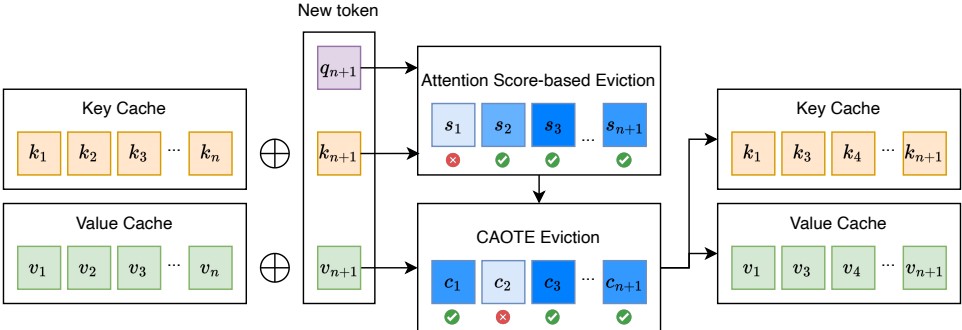

Figure 1: General flow of cache eviction when *CAOTE* is integrated with existing cache eviction methods. In scores part $\{s_1, \ldots s_{n+1}\}$, the lighter color corresponds to smaller score. We show that after including value vectors with eviction scores to get *CAOTE* scores, the token to be evicted can change. Above we changed from evicting token 4 to token 5 after incorporating value vector information.

is inputted and therefore a single token needs to be evicted to maintain the budget $b$. Throughout the paper, we will use eviction error and *CAOTE* score interchangeably.

Given the attention scores of $b+1$ tokens $A = [\alpha_1, \ldots \alpha_{b+1}] \in \mathbb{R}^{1 \times b+1}$ w.r.t. the last input token and the values: $V = [v_1, \ldots, v_{b+1}] \in \mathbb{R}^{d_{\text{head}} \times b+1}$, where $d_{\text{head}}$ is the head dimension. The *CAOTE* score for token $j \in \{1, \ldots, b+1\}$ is defined as (we ignore the layer and head dependence for simplicity).

$$c_j = f_j^{\text{caote}}(A, V) = \frac{\alpha_j}{1 - \alpha_j} |VA^T - v_j|_2 \tag{8}$$

We proof that *CAOTE* score is same as the eviction error. We define eviction error for token $j$ as the mean square error between attention output before and after eviction. Using the same setup as above:

$$e_{\text{eviction},j} = |X_{\text{attn}} - X'_{\text{attn},j}|_2 \tag{9}$$

where, $X_{\text{attn}}$ is attention output before eviction and $X'_{\text{attn},j}$ is attention output after eviction token $j$.

$$X_{\text{attn}} = \alpha_1 v_1 + \alpha_2 v_2 + \ldots \alpha_{b+1} v_{b+1} = VA^T \tag{10}$$

$$X'_{\text{attn},j} = \alpha'_1 v_1 + \ldots \alpha'_{j-1} v_{j-1} + \alpha'_{j+1} v_{j+1} \ldots \alpha'_{b+1} v_{b+1} \tag{11}$$

where, $\alpha'_i \forall i \in \{1, \ldots, j-1, j+1, \ldots b+1\}$ in Eq. (11) is the post-eviction attention score to maintain the sum of the attention score property of sum equal to 1. In the following we show the relation between the pre and post eviction attention score for token $i$ after the eviction of token $j$.

**Theorem 3.1.** *Given a new input token that exceeds the budget (b) by* 1. *A token needs to be evicted. For any token $j$ being evicted, given the retention scores pre-eviction and post-eviction for any token $i \neq j$ as $\alpha_i$ and $\alpha'_i$ respectively, then the following relation holds:*

$$\alpha'_i = \frac{\alpha_i}{1 - \alpha_j} \tag{12}$$

*Proof.* Let the last input token has index $n$, then we define

$$S \triangleq \sum_{l=1}^{n} \exp(q_n^T k_l), \text{ and } S'_j \triangleq S - \exp(q_n^T k_j) \tag{13}$$

The retention score for token $i$ after evicting token $j$ is

$$\alpha'_i = \frac{\exp(q_n^T k_i)}{S'_j} = \frac{\exp(q_n^T k_i)}{S - \exp(q_n^T k_j)} = \frac{\alpha_i}{1 - \alpha_j} \tag{14}$$

$\square$

**Theorem 3.2.** *During generation, the next generated token is inputted back into the model exceeding the budget (b) by* $1$, *invoking token eviction for a single token. For any token* $j$ *that is evicted, the eviction error from Eq.* (9) *and CAOTE score from Eq.* (8) *are exactly same:*

$$c_j = e_{eviction,j} \tag{15}$$

*Proof.* Using Theorem 3.1, we can rewrite post-eviction attention output from Eq. (11)

$$X'_{\text{attn},j} = \frac{1}{1 - \alpha_j} \left( \alpha_1 v_1 + \cdots + \alpha_{j-1} v_{j-1} + \alpha_{j+1} v_{j+1} + \ldots \alpha_{b+1} v_{b+1} \right) \tag{16}$$

$$= \frac{1}{1 - \alpha_j} \left( X_{\text{attn}} - \alpha_j v_j \right) \tag{17}$$

Replacing Eq. (17) in Eq. (9), we get

$$\begin{aligned} e_{\text{eviction},j} &= |X_{\text{attn}} - X'_{\text{attn},j}|_2 \\ &= \frac{\alpha_j}{1 - \alpha_j} |v_j - X_{\text{attn}}|_2 = \frac{\alpha_j}{1 - \alpha_j} |V A^T - v_j|_2 \\ &= c_j \end{aligned} \tag{18}$$

Hence proved. $\qquad\square$

Using Eq. (18) *CAOTE* scores (or eviction error) for each token can be computed in parallel as the dependency is only on attention scores and value vectors. Note that this is the first formulation which that seamlessly integrates attention scores and value vectors into a single score. Any norm can be used for computing *CAOTE* score and based on empirical results we choose $L_2$-norm. In Appendix B.2 we further show that eviction error leads to deviations in downstream task performance due to error in final sampling distribution (or logits), therefore, optimizing for eviction error would also result in less deviation from dense model performance.

## 3.2   CAOTE with general score-based eviction methods

The *CAOTE* formulation allows the use of arbitrary scoring-based eviction methods to incorporate the values into their scoring mechanism, provided that the scores sum to 1.0. In practice, we can adjust the raw eviction scores without changing their relative order by simple normalizations (affine transformations). Let $H$ be the set of retention scores and $f^{\text{norm}}$ be the normalizing function. The *CAOTE* score for general eviction methods is given by:

$$c_j = f_j^{\text{caote}}(f^{\text{norm}}(H), V) \tag{19}$$

$$= \frac{h_j^{\text{norm}}}{1 - h_j^{\text{norm}}} |V(H^{\text{norm}})^T - v_j|_2 \tag{20}$$

where, $h_j^{\text{norm}} = f_j^{\text{norm}}(H)$. We further discuss the generalization of *CAOTE* to well-known token eviction methods in the following.

**CAOTE for H2O**   We consider *H2O* (8), where the scores ($H = [h_1, \ldots, h_{b+1}]$) are based on the sum of previous attention scores, leading to $\Sigma_{j=1}^{b+1} h_j > 1$ during generation-phase as proved in Theorem B.1 in Appendix B.1. In this case, simply dividing each token score by the sum of all scores maps the scores to the range $[0, 1]$ and ensures that new scores follow $\sum_{i=1}^{b+1} h_i^{\text{norm}} = 1$.

$$h_j^{\text{norm}} = \frac{h_j}{\Sigma_{i=1}^{b+1} h_i} \tag{21}$$

For recent methods where all the scores are $\geq 0$, simply dividing by sum of all scores suffices. Note that for *TOVA* (13), this summation is by default equal to $1$.

### 3.3 Fast CAOTE Computation

We also propose a compute-efficient version of *CAOTE*, *FastCAOTE*, with negligible performance degradation and reduced computation by an order of $\frac{1}{d_{\text{hidden}}}$, where $d_{\text{hidden}}$ is the hidden dimension of the model. Here, the pre-eviction attention output ($X_{\text{attn}}$) is replaced with mean of values while everything else remains same, that is, *CAOTE* score for token $j$ is:

$$c_j = \frac{\alpha_j}{1 - \alpha_j}\left|\frac{1}{b+1}\Sigma_{i=1}^{b+1}v_i - v_j\right|_2 \tag{22}$$

## 4 Results

In this section, we demonstrate the efficacy of *CAOTE* for boosting performance on state-of-the-art token eviction methods on a wide range of downstream benchmarks. All experiments were run using Nvidia A100 GPUs.

### 4.1 Experiment Setup

**Tasks** We study the impact of *CAOTE* on different token eviction methods by evaluating on LongBench (17), covering single QA, multiple QA, single/multi-document summarization, synthetic, and code generation tasks. We measure long-context perplexity on the Booksum dataset (18), and lastly, measure recall accuracy on Needle In A Haystack task (19; 20).

**Baselines** We compare the performance of CAOTE to various token eviction methods including: *H2O* (8), *TOVA* (13), and *SnapKV* (14), on LLAMA3 models: Llama 3.2-3B-Instruct and Llama 3.1-8B-Instruct (21), and QWEN2.5 models: Qwen 2.5-3B-Instruct and Qwen 2.5-7B-Instruct (22) for all subsequent experiments.

**Budgets** We evaluated all methods with various KV cache budget sizes of 2048, 4096, 6144, and 8192, denoted by 2k, 4k, 6k, and 8k, respectively.

**Prompt consumption** Unlike other token eviction methods that assume to prefill prompt at once followed by KV cache eviction, we propose to consume tokens in block-wise manner as described in Section 2 with the block-size of 128, i.e., at each inference of LLM during the prefill phase, there are 128 new tokens incoming and being added to the cache, and 128 tokens from the cache are evicted once the total number of tokens reaches the cache budget limit.

### 4.2 LongBench

We present the accuracy of Llama 3.1-8B-Instruct, Llama 3.2-3B-Instruct and, Qwen 2.5-3B-Instruct, Qwen 2.5-7B-Instruct using baseline eviction methods with budget of 2k, 4k, both with and without CAOTE in Table 2 and Table 3. We observe that the best performance is given by *SnapKV-FastCAOTE* for the Llama3 models, while for Qwen 2.5 models *SnapKV-CAOTE* performs the best. *H2O* shows $> 30\%$ improvement with *CAOTE*, while *TOVA, SnapKV* also show overall improvements, making their average accuracy closer to dense accuracy. Additional results for the 6k, 8k budget are shown in Table 6, Table 7 for Llama3 and Qwen 2.5 respectively, in Appendix C.1, which follow a trend similar to the 2k, 4k budgets.

### 4.3 Perplexity

We use the Booksum dataset (18) to measure generation perplexity of different eviction methods for various budgets. In Table 4, we show perplexity gap between a model using a given eviction strategy and that of the model without token eviction with cache budgets of 2k, 4k and 6k. We observe that when *CAOTE* is applied to existing eviction methods, the perplexity either improves or surpasses the perplexity of the baseline model. *TOVA-FastCAOTE*, *SnapKV-CAOTE*, and *SnapKV-FastCAOTE* perform best for 6k, 4k, 2k budgets, respectively, for Llama 3.1-8B-Instruct; for Llama 3.2-3B-Instruct, *TOVA-FastCAOTE* performs best with 2k and 4k budgets and *SnapKV-FastCAOTE* beats other methods using 6k and 8k. Perplexity results for Qwen 2.5 models are shown in Table 8 in Appendix C.2.

Table 2: **LongBench results for Llama 3.1-8B and Llama 3.2-3B-Instruct.** Higher number is better. We highlight the best performing methods within a given budget with **bold** and the second best with underline.

| | | Single Doc. QA | | | Multi Doc. QA | | | Summarization | | | Fewshot Learning | | | Synthetic | | Code | | |
|---|---|---|---|---|---|---|---|---|---|---|---|---|---|---|---|---|---|---|
| | | Narrative QA | Qasper | MF-en | HotpotQA | 2WikiMQA | Musique | GovReport | QMSum | MultiNews | TREC | TriviaQA | SAMSum | PCount | PR-en | Lcc | RB-P | Avg. |
| | Llama 3.1-8B | 30.05 | 47.00 | 56.12 | 57.33 | 47.81 | 32.25 | 34.86 | 25.32 | 27.02 | 73.00 | 91.61 | 43.37 | 8.33 | 99.50 | 61.66 | 51.94 | 49.20 |
| 2k | H2O | 1.74 | 21.15 | 25.33 | 26.11 | 24.15 | 8.78 | 2.17 | 2.70 | 16.78 | 44.00 | 29.36 | 7.62 | 2.25 | 5.88 | 40.15 | 12.14 | 16.89 |
| | + CAOTE | 14.32 | 38.34 | 45.97 | 37.77 | 42.51 | 22.06 | 29.57 | 15.11 | 27.02 | 62.00 | 63.60 | 27.34 | 2.00 | 15.50 | 56.99 | 32.87 | 33.31 |
| | + FastCAOTE | 15.15 | 41.27 | 46.6 | 39.91 | 40.02 | 24.55 | 30.05 | 16.19 | 26.95 | 63 | 62.39 | 26.86 | 3.08 | 17.5 | 56.87 | 34.75 | 34.07 |
| | TOVA | 22.57 | 37.26 | 39.43 | 45.74 | 34.48 | 14.77 | 28.87 | 21.17 | 26.95 | 62.50 | 90.73 | 42.74 | 0.00 | 18.00 | 62.68 | 52.48 | 37.52 |
| | + CAOTE | 21.92 | 37.47 | 38.28 | 45.88 | 35.2 | 15 | 29.02 | 21.21 | 27 | 62.5 | 91.34 | 43.22 | 1.5 | 23 | 62.6 | 54.13 | 38.08 |
| | + FastCAOTE | 21.94 | 38.22 | 38.22 | 46.72 | 36.93 | 14.31 | 29.06 | 21.72 | 26.98 | 63 | 91.65 | 43.53 | 1.5 | 22 | 62.44 | 52.88 | 38.19 |
| | SnapKV | 21.81 | 37.22 | 37.19 | 46.10 | 35.42 | 16.53 | 29.83 | 21.05 | 26.77 | 61.00 | 88.84 | 42.56 | 4.03 | 51.50 | 62.37 | 51.45 | 39.60 |
| | + CAOTE | 21.75 | 37.49 | 36.86 | 44.62 | 37.26 | 16.82 | 30.3 | 21.67 | 26.88 | 64 | 90.65 | 42.8 | 2.09 | 53 | 62.5 | 52.09 | 40.05 |
| | + FastCAOTE | 23.26 | 38.54 | 39.16 | 43.2 | 38.27 | 17.54 | 30.28 | 21.97 | 26.76 | 65.5 | 90.91 | 42.71 | 2.84 | 56 | 62.36 | 52.4 | **40.73** |
| 4k | H2O | 4.07 | 36.16 | 36.00 | 33.52 | 32.87 | 17.78 | 6.66 | 5.95 | 24.09 | 55.00 | 47.65 | 17.41 | 4.00 | 24.50 | 54.85 | 21.43 | 26.37 |
| | + CAOTE | 20.28 | 46.08 | 51.45 | 47.38 | 46.05 | 30.89 | 33.39 | 20.8 | 26.93 | 69 | 80.12 | 38.27 | 4.31 | 32 | 59.22 | 40.51 | 40.42 |
| | + FastCAOTE | 24.4 | 44.32 | 48.11 | 48.19 | 43.69 | 21.12 | 31.55 | 22.36 | 26.98 | 65 | 91.18 | 43.11 | 2 | 46.5 | 61.62 | 53.35 | 42.09 |
| | TOVA | 22.68 | 44.55 | 47.87 | 46.76 | 44.54 | 20.56 | 30.95 | 22.13 | 26.96 | 61.50 | 90.56 | 43.27 | 3.00 | 43.50 | 61.62 | 53.40 | 41.49 |
| | + CAOTE | 24.68 | 43.88 | 48.07 | 49.64 | 44.91 | 22.57 | 31.25 | 22.55 | 26.98 | 63 | 91.29 | 43.29 | 2.5 | 46.5 | 61.6 | 53.45 | 42.24 |
| | + FastCAOTE | 24.4 | 44.32 | 48.11 | 48.19 | 43.69 | 21.12 | 31.55 | 22.36 | 26.98 | 65 | 91.18 | 43.11 | 2 | 46.5 | 61.62 | 53.35 | 42.09 |
| | SnapKV | 24.79 | 44.22 | 47.30 | 48.49 | 46.73 | 20.55 | 32.19 | 22.00 | 26.95 | 67.50 | 90.90 | 43.14 | 5.17 | 89.50 | 61.44 | 51.20 | 45.18 |
| | + CAOTE | 24.41 | 43.16 | 47.77 | 50.87 | 44.11 | 21.04 | 32.51 | 22.98 | 26.93 | 69 | 91.31 | 43.18 | 3.33 | 92 | 61.04 | 51.74 | 45.34 |
| | + FastCAOTE | 24.12 | 44.59 | 47.39 | 50.82 | 44.07 | 22.38 | 32.33 | 22.92 | 27.01 | 69 | 91.31 | 43.53 | 4.58 | 94.5 | 61.31 | 52.11 | **45.75** |
| | Llama 3.2-3B | 23.76 | 40.23 | 50.09 | 50.69 | 42.29 | 26.84 | 33.09 | 24.30 | 25.21 | 72.50 | 90.11 | 42.58 | 3.00 | 96.50 | 56.22 | 56.52 | 45.87 |
| 2k | H2O | 1.63 | 19.96 | 20.20 | 18.02 | 19.56 | 2.88 | 0.78 | 1.55 | 15.97 | 41.00 | 21.97 | 9.83 | 0.50 | 0.50 | 39.71 | 13.91 | 14.25 |
| | + CAOTE | 6.38 | 34.36 | 40.6 | 32.52 | 31.08 | 12.69 | 27.36 | 15.04 | 24.6 | 59 | 52.83 | 26.78 | 3.7 | 7.56 | 51.09 | 36.33 | 28.87 |
| | + FastCAOTE | 7.27 | 34.23 | 39.74 | 32.22 | 30.08 | 12.63 | 27.86 | 15.48 | 25.15 | 60.5 | 53.09 | 26.94 | 2.17 | 8.12 | 51.2 | 35.06 | 28.86 |
| | TOVA | 17.14 | 30.14 | 32.44 | 35.96 | 30.05 | 13.08 | 26.15 | 19.70 | 25.04 | 56.50 | 87.81 | 40.48 | 2.50 | 11.50 | 55.51 | 52.36 | 33.52 |
| | + CAOTE | 17.75 | 30.45 | 32.19 | 37.53 | 29.35 | 13.33 | 26.92 | 19.93 | 25.18 | 60.5 | 88.08 | 41.65 | 1.00 | 12.5 | 54.92 | 53.22 | 34.03 |
| | + FastCAOTE | 17.93 | 30.52 | 33.1 | 37.01 | 30.7 | 13.88 | 26.39 | 20.28 | 24.96 | 60.5 | 88.95 | 41.27 | 2.00 | 12.5 | 55.65 | 53.56 | 34.33 |
| | SnapKV | 17.38 | 31.37 | 31.48 | 37.77 | 30.05 | 11.54 | 27.03 | 19.93 | 24.97 | 59.00 | 88.13 | 40.48 | 3.50 | 32.50 | 56.32 | 55.91 | 35.46 |
| | +CAOTE | 19.11 | 33.12 | 31.09 | 38.68 | 32.09 | 12.31 | 27.48 | 20.38 | 25.2 | 64 | 87.7 | 40.78 | 2.5 | 35 | 57.03 | 56.21 | 36.42 |
| | +FastCAOTE | 18.96 | 32.97 | 33.61 | 39.00 | 31.36 | 12.35 | 27.48 | 20.15 | 25.24 | 65 | 87.2 | 40.7 | 4.5 | 36.5 | 56.06 | 57.12 | **36.76** |
| 4k | H2O | 2.92 | 31.94 | 33.23 | 24.49 | 28.08 | 7.55 | 5.44 | 6.30 | 22.77 | 53.00 | 38.85 | 20.33 | 1.50 | 7.50 | 51.23 | 22.94 | 22.38 |
| | +CAOTE | 12.87 | 40.79 | 47.56 | 40.28 | 39.07 | 16.61 | 30.82 | 19.65 | 25.12 | 65.5 | 69.29 | 34.16 | 2.35 | 17 | 55.32 | 45.12 | 35.09 |
| | +FastCAOTE | 11.85 | 40.41 | 47.93 | 40.81 | 38.93 | 17.36 | 31.22 | 19.67 | 25.1 | 65 | 71.25 | 34.89 | 3.5 | 15 | 55.5 | 44.3 | 35.17 |
| | TOVA | 20.52 | 39.53 | 42.47 | 44.12 | 38.42 | 18.22 | 29.36 | 21.36 | 24.96 | 63.50 | 88.98 | 41.50 | 3.00 | 23.50 | 55.72 | 56.66 | 38.24 |
| | +CAOTE | 19.59 | 39.79 | 42.03 | 45.25 | 37.07 | 19.3 | 29.39 | 21.57 | 24.92 | 63 | 89.14 | 41.77 | 3.00 | 29.5 | 55.68 | 56.19 | 38.57 |
| | +FastCAOTE | 19.77 | 39.23 | 43.13 | 45.28 | 37.04 | 18.82 | 29.25 | 21.94 | 24.96 | 63 | 88.64 | 41.92 | 3.5 | 29 | 55.68 | 56.41 | 38.6 |
| | SnapKV | 19.85 | 39.22 | 39.86 | 45.70 | 37.98 | 16.64 | 29.79 | 21.21 | 25.01 | 65.50 | 89.35 | 40.95 | 2.50 | 62.50 | 55.74 | 56.88 | 40.60 |
| | +CAOTE | 20.1 | 39.74 | 41.01 | 45.64 | 38.26 | 18.64 | 30.07 | 21.53 | 24.98 | 67.5 | 89.08 | 41.78 | 3.00 | 67 | 55.73 | 56.71 | 41.30 |
| | +FastCAOTE | 19.68 | 39.24 | 41.03 | 44.52 | 39.09 | 18.62 | 30.15 | 21.72 | 24.97 | 67 | 88.86 | 41.24 | 3.00 | 71 | 55.67 | 56.64 | **41.40** |

Table 3: **LongBench results for Qwen 2.5-7B/2.5-3B-Instruct.** Higher number is better. We highlight the best performing methods within a given budget with **bold** and the second best with underline.

| | | Single Doc. QA | | | Multi Doc. QA | | | Summarization | | | Fewshot Learning | | | Synthetic | | Code | | |
|---|---|---|---|---|---|---|---|---|---|---|---|---|---|---|---|---|---|---|
| | | Narrative QA | Qasper | MF-en | HotpotQA | 2WikiMQA | Musique | GovReport | QMSum | MultiNews | TREC | TriviaQA | SAMSum | PCount | PR-en | Lcc | RB-P | Avg. |
| | Qwen 2.5-7B | 15.75 | 16.94 | 32.38 | 11.89 | 11.88 | 7.95 | 34.33 | 19.91 | 22.67 | 65.5 | 87.05 | 44.75 | 4.22 | 93.08 | 57.74 | 61.84 | 36.74 |
| 2k | H2O | 2.39 | 7.29 | 12.42 | 8.55 | 11.06 | 2.73 | 3.62 | 6.6 | 15.69 | 42.5 | 28.21 | 10.63 | 0.65 | 0 | 35.1 | 18.77 | 12.89 |
| | + CAOTE | 4.55 | 14.3 | 27.58 | 11.33 | 13.55 | 7.76 | 26.65 | 15.62 | 22.93 | 57 | 49.78 | 27.74 | 1.54 | 11.08 | 51.45 | 32.7 | 23.47 |
| | + FastCAOTE | 4.8 | 12.79 | 28.72 | 12.94 | 13.25 | 7.53 | 27.06 | 14.46 | 22.84 | 59 | 48.23 | 26.4 | 2.53 | 11.54 | 52.85 | 32.93 | 23.62 |
| | TOVA | 8.49 | 14.01 | 21.04 | 14 | 11.51 | 5.09 | 27.43 | 17.84 | 22.83 | 56.5 | 79.56 | 40.55 | 2.43 | 9.29 | 55.99 | 56.15 | 27.67 |
| | + CAOTE | 10.46 | 14.82 | 25.06 | 14.62 | 11.73 | 6.01 | 27.66 | 18.02 | 22.78 | 57.5 | 79.39 | 40.87 | 2.5 | 11.25 | 56.22 | 56.51 | 28.46 |
| | + FastCAOTE | 10.08 | 13.58 | 25.28 | 14.44 | 12.14 | 5.24 | 27.34 | 18.31 | 23.11 | 55.5 | 78.51 | 41.67 | 2.7 | 10.54 | 56.56 | 58.05 | 28.32 |
| | SnapKV | 11.6 | 12.45 | 23.66 | 12.38 | 10.64 | 7.03 | 27.57 | 18.27 | 22.85 | 58 | 81.78 | 41.13 | 3.76 | 19.42 | 55.83 | 56.53 | 28.93 |
| | + CAOTE | 14.02 | 12.23 | 24.55 | 16.45 | 10.35 | 8.59 | 27.77 | 18.91 | 22.87 | 56 | 80.58 | 40.43 | 2.38 | 21.52 | 55.17 | 56.03 | 29.24 |
| | + FastCAOTE | 14.26 | 14.11 | 24.11 | 15.31 | 11.35 | 7.88 | 27.95 | 18.86 | 22.74 | 56.5 | 80.92 | 41.49 | 3.8 | 22.42 | 55.89 | 57.43 | **29.69** |
| 4k | H2O | 1.99 | 11.92 | 19.88 | 10.24 | 10.12 | 4.73 | 9.08 | 10.14 | 20.85 | 51.00 | 37.37 | 20.57 | 3.16 | 6.43 | 52.14 | 29.09 | 18.67 |
| | + CAOTE | 4.78 | 18.06 | 32.49 | 16.23 | 17.28 | 9.57 | 29.81 | 18.04 | 22.86 | 59.5 | 63.05 | 36.91 | 2.7 | 28.25 | 55.13 | 42.42 | 28.57 |
| | + FastCAOTE | 5.69 | 16.99 | 32.63 | 18.22 | 16.58 | 10.48 | 30.3 | 17.71 | 22.88 | 59.5 | 62.95 | 36.29 | 2.1 | 27.65 | 56.3 | 40.65 | 28.56 |
| | TOVA | 12.83 | 17.03 | 27.01 | 16.8 | 13.37 | 8.05 | 29.21 | 19.05 | 22.73 | 58.5 | 82.67 | 42.71 | 1.67 | 15 | 56.69 | 56.59 | 29.99 |
| | + CAOTE | 12.97 | 14.99 | 27.53 | 17.94 | 12.93 | 9.21 | 29.76 | 19.7 | 22.92 | 58 | 82.03 | 43.14 | 2.15 | 17.25 | 57.32 | 59.37 | 30.98 |
| | + FastCAOTE | 14.52 | 16.71 | 26.97 | 18.73 | 13.84 | 9.59 | 29.47 | 19.45 | 22.87 | 59.5 | 82.96 | 42.42 | 2.6 | 20.33 | 57.22 | 58.42 | 30.98 |
| | SnapKV | 14.35 | 13.45 | 28.28 | 16.33 | 11.74 | 8.12 | 29.71 | 19.18 | 22.82 | 57 | 83.8 | 43.27 | 2.41 | 39.83 | 58.12 | 58.67 | 31.69 |
| | + CAOTE | 15.07 | 14.34 | 27.6 | 16.7 | 12.89 | 10.54 | 30.03 | 19.58 | 22.73 | 59.5 | 83.12 | 42.56 | 3.17 | 55.92 | 57.34 | 58.85 | **33.19** |
| | + FastCAOTE | 17.12 | 14.69 | 27.6 | 17.52 | 13.69 | 9.96 | 30.24 | 20.02 | 22.88 | 58.5 | 81.13 | 42.31 | 4.06 | 53.33 | 57.51 | 58.77 | 33.08 |
| | Qwen 2.5-3B | 18.08 | 22.49 | 39.72 | 27.86 | 20.45 | 18.93 | 32.8 | 23.74 | 24.89 | 67.5 | 85.05 | 43.88 | 5 | 40.97 | 51.91 | 47.53 | 35.68 |
| 2k | H2O | 1.8 | 9.18 | 11.62 | 8.54 | 7.31 | 2.77 | 5.93 | 6.99 | 16.89 | 38 | 21.87 | 7.69 | 1 | 3 | 37.36 | 22.9 | 12.68 |
| | + CAOTE | 6.9 | 22.71 | 28.09 | 15.23 | 18.19 | 4.95 | 29.53 | 17.68 | 24.74 | 52.5 | 45.81 | 26.95 | 1.92 | 6.16 | 45.81 | 36.48 | 23.98 |
| | + FastCAOTE | 7.03 | 22.37 | 28.88 | 15.34 | 16.95 | 5.19 | 29.13 | 18.06 | 25.03 | 54.5 | 46.35 | 25.35 | 2.23 | 7.22 | 45.7 | 36.59 | 24.12 |
| | TOVA | 11.69 | 14.94 | 25.33 | 17.29 | 12.58 | 5.91 | 26.67 | 21.49 | 24.78 | 51.5 | 68.4 | 41.79 | 0.23 | 6 | 49.79 | 48.6 | 26.71 |
| | + CAOTE | 11.17 | 15.23 | 27.42 | 18.94 | 13.1 | 6.94 | 27.01 | 21.62 | 24.86 | 57.5 | 68.38 | 42.11 | 0.82 | 4.88 | 49.36 | 48.13 | 27.34 |
| | + FastCAOTE | 11.04 | 15.36 | 27.72 | 19.8 | 13.65 | 6.37 | 27.17 | 22.08 | 24.64 | 57 | 69.13 | 42.48 | 0.77 | 5.25 | 48.36 | 48.58 | 27.46 |
| | SnapKV | 11.7 | 13.91 | 24.28 | 14.8 | 10.89 | 7.42 | 27.4 | 21.63 | 24.64 | 54.5 | 75.34 | 42.72 | 2.5 | 18.33 | 49.65 | 50.59 | 28.14 |
| | +CAOTE | 12.69 | 14.88 | 26.16 | 13.93 | 12.21 | 7.07 | 27.48 | 20.99 | 24.75 | 61 | 75.58 | 42.08 | 4 | 21.29 | 49.94 | 52.38 | **29.15** |
| | +FastCAOTE | 12.03 | 14.56 | 24.82 | 14.66 | 10.83 | 7.89 | 27.51 | 20.83 | 24.64 | 62.5 | 75.51 | 41.53 | 2 | 17 | 49.02 | 50.83 | 28.52 |
| 4k | H2O | 2.82 | 17.34 | 23.27 | 10.18 | 10.47 | 3.03 | 11.06 | 10.73 | 22.93 | 50.75 | 34.93 | 18.03 | 4.35 | 7.32 | 47.74 | 29.42 | 19.02 |
| | +CAOTE | 7.63 | 24.16 | 35.29 | 20.17 | 17.67 | 12.61 | 31.14 | 19.04 | 25.01 | 62.5 | 64.84 | 34.19 | 4.25 | 18.37 | 49.79 | 41.45 | 29.26 |
| | +FastCAOTE | 8.58 | 23.45 | 33.14 | 21.72 | 16.11 | 12.26 | 31.11 | 19.06 | 25.04 | 62 | 65.01 | 35.15 | 4.6 | 17.88 | 50.05 | 40.03 | 29.12 |
| | TOVA | 12.19 | 18.31 | 32.56 | 20.58 | 13.8 | 7.74 | 28.82 | 22.27 | 24.98 | 59 | 80.66 | 43.05 | 1.11 | 9.56 | 49.93 | 46.74 | 29.46 |
| | +CAOTE | 13.16 | 18.67 | 30.74 | 19.33 | 15.7 | 7.32 | 28.93 | 22.14 | 24.91 | 59.5 | 78.54 | 43.57 | 1.55 | 8.25 | 49.4 | 47.31 | 29.31 |
| | +FastCAOTE | 12.2 | 18.55 | 32.29 | 19.17 | 15.13 | 7.23 | 29.12 | 22.44 | 24.97 | 60 | 78.8 | 43.12 | 1.5 | 10.25 | 49.6 | 47.74 | 29.52 |
| | SnapKV | 12.98 | 2.21 | 31.77 | 18.33 | 14.41 | 10.83 | 29.14 | 22.38 | 24.89 | 61 | 84.17 | 42.63 | 3.75 | 25.42 | 50.22 | 48.77 | 30.18 |
| | +CAOTE | 13.65 | 20.35 | 32.62 | 19.36 | 15.27 | 11.42 | 29.47 | 22.44 | 24.78 | 64 | 82.6 | 43 | 4.25 | 24.46 | 50.37 | 49.28 | **31.71** |
| | +FastCAOTE | 13.46 | 19.92 | 32.53 | 20.44 | 13.64 | 8.44 | 29.56 | 22.14 | 24.93 | 64.5 | 82.73 | 43.26 | 2 | 24.58 | 50.14 | 50.08 | 31.40 |

Table 4: **Perplexity difference between different eviction methods with dense baseline.** The lower is better. Negative entry in table means the method performs better than dense baseline. The PPL of Llama 3.2-3B-Instruct and Llama 3.1-8B-Instruct is 15.4911 and 9.833 respectively.

| Budget | H2O | +CAOTE | +FastCAOTE | TOVA | +CAOTE | +FastCAOTE | SnapKV | +CAOTE | +FastCAOTE |
|---|---|---|---|---|---|---|---|---|---|
| | | | | Llama 3.1-8B-Instruct | | | | | |
| 2k | 2.007 | 1.884 | 1.891 | -0.046 | -0.088 | -0.085 | -0.019 | -0.097 | **-0.098** |
| 4k | 1.284 | 1.079 | 1.061 | -0.047 | -0.060 | -0.058 | -0.0483 | **-0.080** | -0.079 |
| 6k | 0.843 | 0.716 | 0.703 | -0.035 | -0.0366 | **-0.085** | -0.036 | -0.043 | -0.045 |
| | | | | Llama 3.2-3B-Instruct | | | | | |
| 2k | 3.814 | 3.561 | 3.563 | 0.493 | 0.442 | **0.432** | 0.555 | 0.451 | 0.435 |
| 4k | 2.460 | 2.142 | 2.128 | 0.175 | 0.150 | **0.144** | 0.223 | 0.152 | 0.144 |
| 6k | 1.369 | 1.219 | 1.187 | 0.065 | 0.057 | 0.057 | 0.076 | 0.056 | **0.044** |
| 8k | 0.589 | 0.462 | 0.448 | 0.023 | 0.012 | 0.011 | 0.020 | 0.007 | **0.005** |

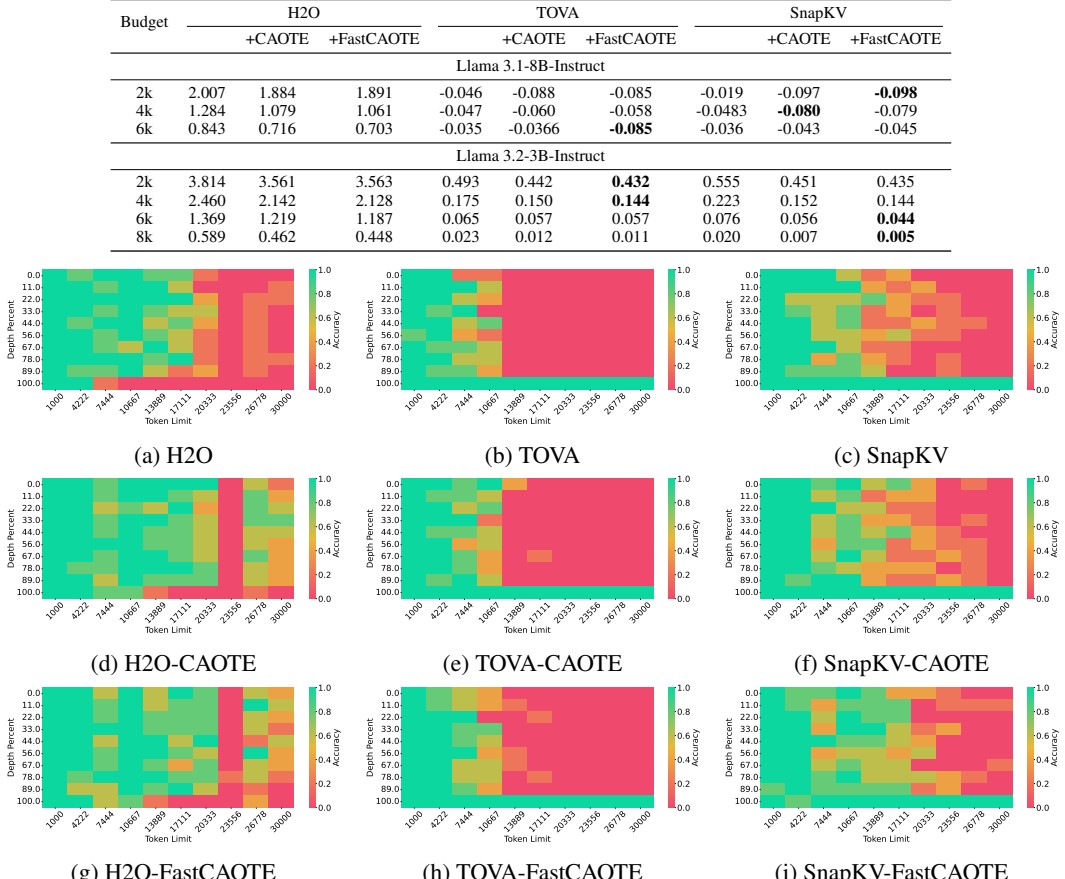

(a) H2O  (b) TOVA  (c) SnapKV

(d) H2O-CAOTE  (e) TOVA-CAOTE  (f) SnapKV-CAOTE

(g) H2O-FastCAOTE  (h) TOVA-FastCAOTE  (i) SnapKV-FastCAOTE

Figure 2: Needle-In-A-Haystack accuracies of Llama 3.1-8B-Instruct with token eviction with 6k cache budget.

## 4.4 Needle In A HayStack

Lastly, we run extensive experiments on Needle-In-A-Haystack benchmark (19; 20) and show quantitative results in Table 5 and visualizations for 6k budget Llama 3.1-8B model in Figure 2. We observe in Table 5 that *H2O-FastCAOTE* performs best for all budgets with Llama 3.2-3B. When using a budget 4k with Llama 3.1-8B, *CAOTE* boosted *H2O* outperforms *TOVA, SnapKV* as well. *H2O-CAOTE* performs best for Llama 3.1-8B with budget = {2k, 6k} and *H2O-FastCAOTE* performs best for 4k budget for Llama 3.1-8b. The gains in precision are especially high for the 4 k budget for the Llama 3.1-8B model, with an increase of up to $30 - 60\%$. We can see in Figure 2 that *CAOTE* improves the state-of-the-art eviction method and is able to predict beyond their budget constraints. Results for Qwen 2.5 models are shown in Table 9 in Appendix C.3.

## 5 Related Work

**Sparse and Efficient Attention** Sparse or efficient attention based methods result in mitigating the computation load and saving memory consumption by using efficient linear attentions (23). Additionally, there are KV compression methods which don't evict any tokens as post eviction the token is not retrievable, (24) proposes to keep important tokens based on attention score in cache while combining the evicted tokens via linear attention into single embedding. Landmark attention injects learnable special tokens between chunks of tokens and access past tokens in chunks instead of

Table 5: **Needle-in-haystack accuracy** for Llama 3.2-3B/3.1-8B-Instruct using baseline eviction methods with(out) *CAOTE*. Higher is better, maximum accuracy is 1.0.

| Budget | H2O | | +FastCAOTE | TOVA | | +FastCAOTE | SnapKV | | +FastCAOTE |
|---|---|---|---|---|---|---|---|---|---|
| | | +CAOTE | | | +CAOTE | | | +CAOTE | |
| Llama 3.1-8B-Instruct | | | | | | | | | |
| 2k | 0.174 | **0.270** | 0.264 | 0.196 | 0.204 | 0.202 | 0.214 | 0.226 | 0.242 |
| 4k | 0.330 | 0.538 | **0.568** | 0.286 | 0.298 | 0.292 | 0.360 | 0.392 | 0.420 |
| 6k | 0.544 | **0.698** | 0.676 | 0.370 | 0.402 | 0.396 | 0.490 | 0.550 | 0.580 |
| Llama 3.2-3B-Instruct | | | | | | | | | |
| 2k | 0.104 | 0.160 | **0.172** | **0.172** | 0.150 | 0.166 | 0.154 | **0.172** | 0.168 |
| 4k | 0.198 | 0.262 | **0.294** | 0.220 | 0.232 | 0.232 | 0.226 | 0.222 | 0.232 |
| 6k | 0.258 | 0.308 | **0.322** | 0.258 | 0.278 | 0.270 | 0.272 | 0.264 | 0.312 |
| 8k | 0.324 | 0.414 | **0.404** | 0.338 | 0.364 | 0.344 | 0.342 | 0.336 | 0.366 |

individually. Lastly, there are better architectures based with constant KV memory which outperform linear attention attentions (25). However, all these methods require either from-scratch training or fine-tuning.

**KV Cache Eviction**   At the extreme end of efficient KV cache management, token eviction methods have been extensively studied. Leveraging the sparsity of attention in LLMs (10; 26; 27), these methods determine the importance of KV pairs using (learned) rules and retain the pairs with high scores in the cache to approximate the attention output. StreamingLLM (10) observes an attention sink phenomenon, which states that the first few tokens tend to receive the majority of attention weights. To exploit this, it proposes SinkAttention, which prioritizes keeping the initial tokens in the cache while doing sliding window-based attention. Other methods, such as H2O (8), TOVA (13), SnapKV (14), and RoCO (28), retain tokens with high attention scores in the KV cache with various algorithmic modifications. These include preserving the first or last tokens in addition to those with high attention scores or applying smoothing techniques to the attention scores. While these token eviction methods primarily rely on attention scores to assess token importance, CAOTE introduces an orthogonal scoring metric that estimates the impact of values on approximating attention outputs. This metric can complement existing token importance scoring approaches, enhancing other eviction methods.

Furthermore, other lines of work enable layer-wise budget optimization (29) by considering the scores of all heads jointly, and selecting top-K nodes, while others (30) consider managing memory by keeping/discarding based on token characteristics with baseline token eviction (H2O). Our proposed method, CAOTE is highly flexible, and can be integrated with both the mentioned methods to achieve additional boost on performance.

## 6   Conclusion

We propose a post-training KV cache eviction method that can be seamlessly integrated with any existing eviction strategies. Our approach, CAOTE, introduces an optimization objective aimed at minimizing the alteration in attention output when evicting a token. This objective ensures the incorporation of both attention scores and value vectors in the eviction decision process. Our formulation allows for the parallel computation of the CAOTE score for all tokens. Additionally, we present an efficient variant, FastCAOTE. Through extensive evaluations across various downstream tasks, we demonstrate that eviction methods equipped with CAOTE consistently deliver superior performance.

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

# A    Limitation

*CAOTE* is a myopic (greedy) strategy and its scoring framework based on the assumption of evicting 1 token per iteration. This assumption breaks during prefilling stage, however, taking into account change in attention output due to multi-token eviction is non-trivial. Fortunately, we observe that even assuming multi-token eviction independently (without considering the effect of other tokens being evicted), *CAOTE* is still able to give boost in performance for all tasks. Due to this reason *CAOTE*'s performance might further improve with smaller prompt filling block-size.

# B    Additional Proofs

## B.1    H2O scores

**Theorem B.1.** *Given H2O scores for $b + 1$ tokens as $[h_1, \ldots, h_{b+1}]$, the summation of all $h_i$, $\forall i \in \{1, \ldots, b+1\}$ is greater than 1*

$$\Sigma_{i=1}^{b+1} h_i > 1 \tag{23}$$

*Proof.* Assuming that only $b+1$ tokens are present and are propagated through the model at the same time. The causal attention mask $A \in [0, 1]^{b+1 \times b+1}$, will have all entries on upper triangle excluding diagonal is 0. The first token will attend to itself have attention score as 1

$$A_{1,1} = 1 \tag{24}$$
$$A_{1,>1} = 0 \tag{25}$$

H2O score for token 1 is defined as the sum of attention score to token 1 by all future tokens:

$$h_1 = A_{1,1} + A_{2,1} + \cdots + A_{b+1,1} = \Sigma_{i=1}^{b+1} A_{i,1} \tag{26}$$

In general for a token $j$, the H2O score is defined as

$$h_j = A_{j,j} + A_{j+1,j} + \cdots + A_{b+1,j} \tag{27}$$
$$= \underbrace{A_{1,j} + \cdots + A_{j-1,j}}_{=0 \text{ causal mask}} + A_{j,j} \cdots + A_{b+1,j} \tag{28}$$
$$= \Sigma_{i=1}^{b+1} A_{i,j} \tag{29}$$

Using Eq. (29) and summing for all H2O scores, we get

$$\Sigma_{j=1}^{b+1} h_j = \Sigma_{j=1}^{b+1} \Sigma_{i=1}^{b+1} A_{i,j} \tag{30}$$

$$= \Sigma_{i=1}^{b+1} \Sigma_{j=1}^{b+1} A_{i,j} \tag{31}$$

$$= \Sigma_{i=1}^{b+1} ( \underbrace{\Sigma_{j=1}^{i} A_{i,j}}_{=1 \text{ due to softmax}} + \underbrace{\Sigma_{j=j+1}^{b+1} A_{i,j}}_{=0 \text{ for causal mask}} ) \tag{32}$$

$$= b + 1 > 1 \tag{33}$$

Hence proved.

$\square$

## B.2 Relation of CAOTE score (eviction error) to downstream logits

We show that evicting token based on *CAOTE* score can lead to smaller discrepancy in final logits which affect the downstream performance. As *CAOTE* score is the eviction error during generation phase, we instead show the relation between eviction error and logits. We start by showing for a single attention layer (single head) based network, its extension to multiple heads and, finally a transformer layer (self-attention and feed-forward-network). For simplicity we ignore layer-norms. Some definitions which will used are given below. We assume a budget of $b$, with current token sequence having $b + 1$ tokens (superscript denotes layer):

$$X^0 \triangleq [x_1, x_2, \ldots, x_{b+1}], \ X_{b+1}^0 \triangleq x_{b+1} \tag{34}$$

The attention output for a sequence of $b + 1$ tokens is (for layer $m$)

$$X_{A,b+1}^m \triangleq \Sigma_{j=1}^{b+1} \alpha_j^m W_v^m X_j^m \tag{35}$$

The logits with and without eviction for token $j$ are defined as $l_j, \hat{l}_j$ respectively.

**(Case 1) Single self-attention layer with single head**: The logits for dense baseline is:

$$X_{b+1}^1 = X_{b+1}^0 + W_O X_{A,b+1}^0 \tag{36}$$

$$l_{b+1} = W_H X_{b+1}^1 = W_H (X_{b+1}^0 + W_O \Sigma_j \alpha_j W_V X_j^0) \tag{37}$$

where $W_H, W_O, W_V$ are the LM-head, output projection, and value projection respectively. $X_{b+1}^1$ is the output after the residual connection.

The logits with eviction will have a perturbation due to error in attention output (CAOTE score or eviction error), and is given as:

$$\hat{X}_{b+1}^1 = X_{b+1}^0 + W_O X_{A,b+1}^0 + W_O \Delta_A^0 \tag{38}$$

$$\hat{l}_{b+1} = W_H \hat{X}_{b+1}^1 \tag{39}$$

$$= W_H (X_{b+1}^0 + W_O X_{A,b+1} + W_O \Delta_A^0) \tag{40}$$

$$= l_{b+1} + \Delta_{l,b+1} \tag{41}$$

where the logit error is $\Delta_{l,b+1} = W_H W_O \Delta_{A,b+1}, \ \Delta_{A,b+1} = e_{\text{eviction}}$ from (9).

**(Case 2) Multiple attention heads**: this is trivial and can be achieved by replacing $\Delta_A = \text{concat}(\Delta_A^1, \ldots, \Delta_A^h)$, where super-script denotes head number.

**(Case 3) Single self-attention and feedforward-network (FFN)**: we still assume single head without layer-norms. The dense logit is given as

$$X_{b+1}^{1/2} = X_{b+1}^0 + W_O X_{A,b+1}^0 \tag{42}$$

$$X_{b+1}^1 = X_{b+1}^{1/2} + W_{FFN} X_{b+1}^{1/2} \tag{43}$$

$$= X_{b+1}^0 + W_O X_{A,b+1}^0$$
$$+ W_{FFN} X_{b+1}^0 + W_{FFN} W_O X_{A,b+1}^0 \tag{44}$$

$$l_{b+1} = W_H X_{b+1}^1 \tag{45}$$

Table 6: **LongBench results for Llama 3.1-8B and Llama 3.2-3B-Instruct.** Higher number is better. We highlight the best performing methods within a given budget with **bold** and the second best with underline.

| | | Single Doc. QA | | | Multi Doc. QA | | | Summarization | | | Fewshot Learning | | | Synthetic | | Code | | |
|---|---|---|---|---|---|---|---|---|---|---|---|---|---|---|---|---|---|---|
| | | Narrative QA | Qasper | MF-en | HotpotQA | 2WikiMQA | Musique | GovReport | QMSum | MultiNews | TREC | TriviaQA | SAMSum | PCount | PR-en | Lcc | RB-P | Avg. |
| | Llama 3.1-8B | 30.05 | 47.00 | 56.12 | 57.33 | 47.81 | 32.25 | 34.86 | 25.32 | 27.02 | 73.00 | 91.61 | 43.37 | 8.33 | 99.50 | 61.66 | 51.94 | 49.20 |
| 6k | Sink | 25.41 | 47.40 | 44.13 | 47.39 | 45.73 | 21.90 | 32.53 | 22.19 | 26.87 | 72.00 | 91.25 | 43.41 | 3.08 | 52.50 | 62.22 | 56.24 | 43.39 |
| | H2O | 8.52 | 43.31 | 44.80 | 40.03 | 42.46 | 21.68 | 11.85 | 8.78 | 26.03 | 62.00 | 56.39 | 25.72 | 5.75 | 45.50 | 58.62 | 29.53 | 33.19 |
| | + CAOTE | 25.77 | 46.45 | 54.99 | 50.57 | 47.7 | 31.93 | 33.99 | 22.86 | 27.01 | 71 | 87.05 | 41.29 | 6.67 | 54 | 60.48 | 43.23 | 44.06 |
| | + FastCAOTE | 27.02 | 46.46 | 55.4 | 51.32 | 47.4 | 32.89 | 34.08 | 23.69 | 27.03 | 71 | 86.25 | 42.14 | 9 | 48.5 | 60.49 | 42.94 | 44.10 |
| | TOVA | 24.59 | 45.93 | 53.92 | 55.09 | 47.43 | 25.07 | 32.33 | 24.10 | 27.00 | 68.50 | 90.81 | 43.89 | 4.25 | 67.00 | 61.50 | 52.39 | 45.24 |
| | + CAOTE | 24.23 | 45.88 | 54.5 | 52.96 | 49.59 | 27.02 | 32.62 | 23.86 | 27.08 | 70 | 90.98 | 43.45 | 3 | 74.5 | 61.46 | 51.76 | 45.74 |
| | + FastCAOTE | 24.17 | 46.07 | 53.8 | 53.53 | 48.11 | 26.49 | 32.64 | 23.88 | 27.01 | 70 | 90.81 | 43.53 | 3 | 73 | 61.49 | 51.43 | 45.56 |
| | SnapKV | 24.10 | 45.57 | 50.44 | 53.12 | 48.41 | 24.27 | 33.43 | 23.53 | 27.03 | 71.50 | 92.28 | 43.58 | 5.25 | 98.00 | 61.32 | 52.16 | 47.12 |
| | + CAOTE | 25.97 | 46.09 | 51.54 | 55.19 | 47.41 | 26.48 | 33.32 | 24 | 27.05 | 71.5 | 91.11 | 43.55 | 6.83 | 99.5 | 61.11 | 51.45 | 47.63 |
| | + FastCAOTE | 24.77 | 46.06 | 52.18 | 56.72 | 47.01 | 26.24 | 33.41 | 23.8 | 26.99 | 73 | 91.31 | 43.6 | 5.92 | 99.5 | 61.5 | 51.37 | **47.71** |
| 8k | Sink | 23.53 | 46.63 | 48.68 | 49.61 | 47.16 | 21.14 | 33.10 | 23.20 | 26.92 | 72.00 | 91.29 | 43.79 | 3.25 | 66.00 | 62.18 | 56.43 | 44.68 |
| | H2O | 13.85 | 44.94 | 47.81 | 43.64 | 44.90 | 23.65 | 18.78 | 11.35 | 26.49 | 69.50 | 69.05 | 33.41 | 5.25 | 62.50 | 59.74 | 36.26 | 38.20 |
| | + CAOTE | 27.74 | 46.67 | 54.97 | 52.71 | 48.28 | 33.66 | 34.51 | 24.73 | 26.99 | 73 | 86.8 | 42.86 | 5 | 66.5 | 61.06 | 48.29 | 45.86 |
| | + FastCAOTE | 28.8 | 47.08 | 54.67 | 52.95 | 47.13 | 34.36 | 34.21 | 24.53 | 27.04 | 72 | 87.87 | 43.03 | 5.5 | 69.5 | 61.06 | 49.69 | 46.21 |
| | TOVA | 24.86 | 46.78 | 54.83 | 54.52 | 49.00 | 26.40 | 33.44 | 24.76 | 27.00 | 71.00 | 91.11 | 43.29 | 6.25 | 87.00 | 61.49 | 51.79 | 47.09 |
| | + CAOTE | 25.65 | 46.88 | 54.5 | 54.73 | 48.73 | 26.54 | 33.47 | 24.8 | 27.02 | 72 | 91.11 | 43.26 | 6.25 | 87 | 61.36 | 51.3 | 47.21 |
| | + FastCAOTE | 25.25 | 46.75 | 54.76 | 56.29 | 48.94 | 26.25 | 33.37 | 24.81 | 27.01 | 71.5 | 91.11 | 43.24 | 5.25 | 89 | 61.32 | 52.1 | **48.58** |
| | SnapKV | 25.15 | 46.55 | 53.39 | 56.00 | 48.75 | 27.82 | 33.67 | 24.85 | 27.01 | 72.50 | 91.78 | 43.54 | 5.08 | 100.00 | 61.48 | 51.41 | 48.06 |
| | + CAOTE | 27.06 | 46.42 | 53.76 | 56.51 | 47.79 | 28.13 | 33.87 | 24.93 | 27.02 | 73 | 91.38 | 43.26 | 6.75 | 99.5 | 61.49 | 51.99 | 48.31 |
| | + FastCAOTE | 26.91 | 46.59 | 53.47 | 56.63 | 48.54 | 29.27 | 33.91 | 24.86 | 27.01 | 73 | 91.38 | 43.49 | 6.75 | 100 | 61.49 | 51.78 | 48.44 |
| | Llama 3.2-3B | 23.76 | 40.23 | 50.09 | 50.69 | 42.29 | 26.84 | 33.09 | 24.30 | 25.21 | 72.50 | 90.11 | 42.58 | 3.00 | 96.50 | 56.22 | 56.52 | 45.87 |
| 6k | Sink | 19.33 | 40.29 | 37.95 | 46.48 | 40.29 | 15.31 | 30.43 | 21.35 | 25.14 | 71.50 | 88.93 | 42.04 | 3.50 | 47.00 | 56.55 | 54.11 | 40.01 |
| | H2O | 4.62 | 38.81 | 39.06 | 34.66 | 35.52 | 15.21 | 10.51 | 10.01 | 24.25 | 61.50 | 53.23 | 27.37 | 0.50 | 13.00 | 54.55 | 32.29 | 28.44 |
| | +CAOTE | 16.14 | 41.68 | 49.36 | 46.7 | 43.36 | 22.75 | 32.07 | 21 | 25.07 | 69 | 80.02 | 39.33 | 1.5 | 26 | 55.82 | 49.05 | 38.68 |
| | +FastCAOTE | 16.31 | 41.94 | 49.17 | 45.64 | 41.83 | 21.68 | 32.07 | 20.73 | 25.02 | 68.5 | 80.34 | 39.88 | 3.5 | 24 | 55.83 | 48.7 | 38.45 |
| | TOVA | 20.22 | 39.78 | 45.86 | 49.08 | 41.54 | 20.43 | 30.50 | 22.17 | 25.11 | 66.50 | 89.00 | 42.50 | 4.00 | 46.50 | 55.57 | 57.53 | 41.02 |
| | +CAOTE | 21.17 | 39.69 | 47.21 | 48.82 | 41.7 | 20.59 | 30.72 | 22.36 | 25.1 | 68 | 89 | 42.38 | 3.5 | 52.5 | 55.6 | 57.09 | 41.59 |
| | +FastCAOTE | 21.48 | 39.66 | 47.02 | 47.56 | 41.95 | 19.91 | 30.8 | 21.98 | 25.17 | 67.5 | 89.5 | 42.06 | 4 | 53 | 55.6 | 57.39 | 41.54 |
| | SnapKV | 20.83 | 39.65 | 44.48 | 49.30 | 40.18 | 20.28 | 31.27 | 22.73 | 25.09 | 69.00 | 89.95 | 41.47 | 4.00 | 85.00 | 55.69 | 57.82 | 43.55 |
| | +CAOTE | 20.23 | 39.65 | 44.91 | 50.16 | 40.58 | 21.32 | 31.23 | 22.51 | 25.13 | 69 | 90 | 41.83 | 5 | 89.5 | 55.84 | 57.24 | 44.01 |
| | +FastCAOTE | 20.09 | 40.02 | 44.58 | 48.57 | 42.12 | 22.51 | 31.25 | 22.89 | 25.15 | 71 | 90 | 41.83 | 4 | 89.5 | 55.83 | 57.25 | **44.16** |
| 8k | Sink | 20.15 | 40.02 | 41.94 | 48.15 | 42.24 | 16.01 | 31.64 | 22.10 | 25.20 | 73.00 | 89.26 | 42.37 | 3.50 | 62.50 | 56.86 | 56.63 | 41.97 |
| | H2O | 9.65 | 39.66 | 43.20 | 38.09 | 40.41 | 21.46 | 17.80 | 13.28 | 24.67 | 70.00 | 64.30 | 32.19 | 2.00 | 24.50 | 55.00 | 39.09 | 33.46 |
| | +CAOTE | 20.07 | 40.73 | 47.76 | 47.25 | 42.88 | 23.19 | 32.41 | 22.01 | 25.15 | 71 | 83.58 | 40.8 | 3 | 43.5 | 55.45 | 53.35 | 40.76 |
| | +FastCAOTE | 20.81 | 40.54 | 48.1 | 47.35 | 43.4 | 25.13 | 32.73 | 22.31 | 25.13 | 71.5 | 84.91 | 40.6 | 4 | 45 | 55.84 | 52.89 | 41.27 |
| | TOVA | 21.08 | 40.67 | 49.07 | 48.69 | 41.93 | 23.05 | 31.64 | 22.85 | 25.21 | 69.00 | 89.25 | 42.19 | 2.50 | 71.00 | 55.77 | 57.47 | 43.21 |
| | +CAOTE | 21.97 | 40.66 | 49.37 | 50.1 | 41.29 | 24.05 | 31.65 | 22.85 | 25.16 | 69.5 | 89.5 | 42 | 3 | 78.5 | 55.82 | 57.16 | 43.91 |
| | +FastCAOTE | 22.73 | 40.51 | 49.36 | 50.18 | 42.26 | 24.45 | 31.68 | 23.09 | 25.16 | 69.5 | 89.5 | 42.28 | 4.5 | 80 | 55.79 | 57.16 | 44.26 |
| | SnapKV | 20.49 | 40.80 | 48.16 | 48.78 | 41.65 | 24.79 | 31.81 | 23.46 | 25.17 | 70.00 | 90.17 | 41.99 | 5.00 | 94.00 | 55.77 | 57.29 | **44.96** |
| | +CAOTE | 19.71 | 40.7 | 48.05 | 49.03 | 41.27 | 22.95 | 31.95 | 23.1 | 25.21 | 72 | 90 | 41.88 | 4 | 95 | 55.77 | 57.02 | 44.85 |
| | +FastCAOTE | 20.13 | 40.71 | 48.35 | 48.62 | 41.04 | 24.38 | 32.19 | 23.04 | 25.20 | 72 | 90 | 42.33 | 3.5 | 95 | 55.77 | 57.03 | **44.96** |

where, for simplicity we assume feedforward network to subsumed within $W_{FFN}$.

The perturbed logit due to eviction is given as:

$$\hat{X}_{b+1}^{1/2} = X_{b+1}^0 + W_O X_{A,b+1}^0 + W_O \Delta_A^0 \tag{46}$$

$$\hat{X}_{b+1}^1 = \hat{X}_{b+1}^{1/2} + W_{FFN} \hat{X}_{b+1}^{1/2} \tag{47}$$

$$= X_{b+1}^0 + W_O X_{A,b+1}^0 + W_O \Delta_A^0$$
$$+ W_{FFN} X_{b+1}^0 + W_{FFN} W_O X_{A,b+1}^0$$
$$+ W_{FFN} W_O \Delta_A^0 \tag{48}$$

$$\hat{l}_{b+1} = W_H \hat{X}_{b+1}^1 \tag{49}$$

$$= l_{b+1} + \Delta_{l,b+1} \tag{50}$$

where, the logit error $\Delta_{l,b+1} = W_H(W_O \Delta_A^0 + W_{FFN} W_O \Delta_A^0)$. Thus, the above analysis shows that error in attention output can cause discrepancy in logit space which can affect performance on downstream tasks.

Note that for multiple layers, each layer would have its own eviction error which will keep compounding; however, this computing the exact compounded error is non-trivial due to the presence of output layer-norms.

# C  Additional Results

## C.1  LongBench

LongBench result for 6k and 8K for Llama 3.2-3B-Instruct/3.1-8B-Instruct and Qwen 2.5-3B-Instruct/8B-Instruct are shown in Table Table 6, Table 7 respectively. We also include Sink attention

Table 7: **LongBench results for Qwen 2.5-7B/2.5-3B-Instruct.** Higher number is better. We highlight the best performing methods within a given budget with **bold** and the second best with underline.

| | | Single Doc. QA | | | Multi Doc. QA | | | Summarization | | | Fewshot Learning | | | Synthetic | | Code | | |
| --- | --- | --- | --- | --- | --- | --- | --- | --- | --- | --- | --- | --- | --- | --- | --- | --- | --- | --- |
| | | Narrative QA | Qasper | MF-en | HotpotQA | 2WikiMQA | Musique | GovReport | QMSum | MultiNews | TREC | TriviaQA | SAMSum | PCount | PR-en | Lcc | RB-P | Avg. |
| | Qwen 2.5-7B | 15.75 | 16.94 | 32.38 | 11.89 | 11.88 | 7.95 | 34.33 | 19.91 | 22.67 | 65.5 | 87.05 | 44.75 | 4.22 | 93.08 | 57.74 | 61.84 | 36.74 |
| | Sink | 7.37 | 16.61 | 25.73 | 11.29 | 11.27 | 5.69 | 31.47 | 18.72 | 22.86 | 64.5 | 84.86 | 44.47 | 3.59 | 41.48 | 55.89 | 55.99 | 31.36 |
| | H2O | 3.34 | 14.79 | 23.94 | 11.45 | 11.3 | 5.52 | 14.63 | 14.27 | 22.06 | 55.75 | 51.99 | 28.01 | 1.39 | 9.41 | 54.68 | 38.32 | 22.55 |
| | + CAOTE | 4.78 | 18.06 | 32.49 | 16.23 | 17.28 | 9.57 | 29.81 | 18.04 | 22.86 | 59.5 | 63.05 | 36.91 | 2.7 | 28.25 | 55.13 | 42.42 | 28.57 |
| | + FastCAOTE | 5.69 | 16.99 | 32.62 | 18.22 | 16.58 | 10.48 | 30.3 | 17.71 | 22.88 | 59.5 | 62.95 | 36.29 | 2.1 | 27.65 | 56.3 | 40.65 | 28.56 |
| 6k | TOVA | 15.77 | 15.33 | 30.31 | 19.3 | 13.78 | 9.11 | 30.4 | 19.95 | 22.91 | 61.5 | 83.47 | 42.9 | 1.15 | 21.775 | 57.68 | 57.99 | 31.46 |
| | + CAOTE | 15.81 | 16.07 | 29.39 | 19.4 | 14.15 | 10.8 | 30.89 | 20.54 | 22.86 | 62 | 84.92 | 43.19 | 2.17 | 30 | 57.76 | 57.53 | 32.34 |
| | + FastCAOTE | 15.67 | 16.23 | 30.4 | 19.45 | 13.32 | 10.18 | 30.77 | 20.2 | 22.82 | 61.5 | 83.29 | 43.3 | 1.5 | 28.75 | 57.71 | 58.1 | 32.07 |
| | SnapKV | 14.34 | 16.35 | 31.12 | 17.56 | 14.1 | 8.74 | 31.09 | 20.16 | 22.84 | 60 | 83.8 | 42.99 | 2.91 | 54.17 | 57.48 | 60.26 | 33.62 |
| | + CAOTE | 12.77 | 16.3 | 31.33 | 19.74 | 14.06 | 11.07 | 31.02 | 20.85 | 22.91 | 61.5 | 83.79 | 42.97 | 4.8 | 68.25 | 57.54 | 61.08 | **35.00** |
| | + FastCAOTE | 12.98 | 15.93 | 31.3 | 18.58 | 13.82 | 9.45 | 30.96 | 20.27 | 22.88 | 61.5 | 84.58 | 43.28 | 5.34 | 65.48 | 57.54 | 60.25 | 34.63 |
| | H2O | 6.1 | 15.55 | 28.29 | 12.37 | 14.65 | 6.24 | 20.78 | 17.22 | 22.44 | 59 | 58.74 | 33.05 | 1.82 | 15.73 | 55.63 | 44.56 | 25.76 |
| | + CAOTE | 8.65 | 15.59 | 34.92 | 20.41 | 15.95 | 13.6 | 32.11 | 20.05 | 22.82 | 63.5 | 78.2 | 40.66 | 3.85 | 46.33 | 57.19 | 51.7 | 32.85 |
| | + FastCAOTE | 6.76 | 15.88 | 34.3 | 20.75 | 16.2 | 16.82 | 31.95 | 20.67 | 22.81 | 62.5 | 77.33 | 41.02 | 3.03 | 47.08 | 57.23 | 50.48 | 32.8 |
| 8k | TOVA | 15.69 | 15.55 | 33.09 | 18.37 | 13.99 | 11.26 | 31.33 | 20.17 | 22.82 | 62 | 84.49 | 43.01 | 2.78 | 30.33 | 57.45 | 58.96 | 32.58 |
| | + CAOTE | 16.38 | 15.46 | 32.16 | 17.86 | 14.24 | 12.76 | 31.34 | 20.2 | 22.8 | 61 | 83.97 | 43.23 | 2.01 | 38.83 | 57.45 | 59.41 | 33.07 |
| | + FastCAOTE | 17.06 | 15.55 | 32.32 | 17.57 | 14.14 | 13.03 | 31.43 | 20.3 | 22.78 | 62 | 84.86 | 43.3 | 1.75 | 41.58 | 57.41 | 58.93 | 33.42 |
| | SnapKV | 15.6 | 15.81 | 33.47 | 18.02 | 14.49 | 10.53 | 31.99 | 20.09 | 22.84 | 61 | 84.08 | 43.01 | 4.58 | 64.25 | 57.46 | 60.59 | 34.86 |
| | + CAOTE | 15.55 | 15.57 | 33.89 | 21.08 | 14.43 | 12.38 | 31.41 | 20.73 | 22.83 | 61.5 | 85.11 | 43.39 | 5.22 | 75.75 | 57.44 | 60.35 | **36.04** |
| | + FastCAOTE | 13.38 | 15.77 | 33.97 | 19.78 | 15.08 | 13.01 | 31.44 | 20.69 | 22.77 | 62 | 85.66 | 43.69 | 4.24 | 75.33 | 57.44 | 60.04 | 35.89 |
| | Qwen 2.5-3B | 18.08 | 22.49 | 39.72 | 27.86 | 20.45 | 18.93 | 32.8 | 23.74 | 24.89 | 67.5 | 85.05 | 43.88 | 5 | 40.97 | 51.91 | 47.53 | 35.68 |
| | Sink | 13.01 | 20.03 | 32.59 | 18.62 | 15.77 | 9.37 | 30.98 | 20.7 | 24.97 | 66.5 | 75.39 | 42.77 | 4 | 14.92 | 52.32 | 50.35 | 30.77 |
| | H2O | 5.52 | 18.62 | 27.93 | 12.61 | 15.07 | 4.26 | 14.92 | 13.89 | 24.21 | 58 | 45.94 | 24.93 | 2.91 | 9.1 | 49.5 | 34.54 | 22.62 |
| | + CAOTE | 8.23 | 21.34 | 36.28 | 22.43 | 17.92 | 13.53 | 31.57 | 21.2 | 24.91 | 65 | 74.3 | 39.09 | 4.58 | 21.25 | 50.47 | 42.71 | 30.93 |
| | + FastCAOTE | 9.29 | 20.47 | 35.8 | 22.67 | 18.14 | 13.65 | 31.34 | 20.52 | 24.82 | 64.5 | 75.88 | 39.16 | 5.72 | 20.42 | 50.6 | 44.38 | 31.02 |
| 6k | TOVA | 13.62 | 19.56 | 34.64 | 21.67 | 16.25 | 8.47 | 30.17 | 23.1 | 24.94 | 63.5 | 81.88 | 42.97 | 1.16 | 10.58 | 51.3 | 47.7 | 30.72 |
| | + CAOTE | 13.28 | 19.82 | 35.25 | 22.6 | 15.5 | 8.97 | 30.4 | 23.17 | 24.84 | 64.5 | 81.68 | 43.46 | 2.07 | 13.21 | 50.56 | 47.05 | 31.02 |
| | + FastCAOTE | 13.34 | 19.71 | 35.58 | 22.02 | 15.65 | 8.76 | 30.49 | 23.26 | 24.82 | 65 | 80.92 | 43.66 | 1.75 | 13 | 51.45 | 47.74 | 31.07 |
| | SnapKV | 14.16 | 20.09 | 36.15 | 19.14 | 15.59 | 12.7 | 30.35 | 22.75 | 24.91 | 65 | 83.92 | 43.52 | 5.00 | 32.2 | 51.04 | 47.49 | 32.75 |
| | +CAOTE | 14.3 | 20.13 | 34.89 | 20.32 | 15.06 | 12.85 | 30.61 | 23.16 | 24.9 | 66.5 | 84.75 | 43.3 | 4.75 | 33.9 | 51.48 | 48.31 | **33.08** |
| | +FastCAOTE | 14.33 | 19.48 | 35.58 | 20.8 | 16.54 | 11.85 | 30.68 | 23.19 | 24.93 | 66.5 | 83.54 | 43.55 | 4.62 | 33 | 51.06 | 47.93 | 32.91 |
| | H2O | 6.16 | 19.84 | 32.32 | 16.01 | 17.74 | 4.99 | 20.21 | 16.49 | 24.54 | 64 | 56.1 | 32.56 | 3.13 | 11.61 | 50.61 | 38.8 | 25.94 |
| | +CAOTE | 11.53 | 21.59 | 38.02 | 25.62 | 20.19 | 15.11 | 32.18 | 22.81 | 24.81 | 67.5 | 79.32 | 41.2 | 5.15 | 25.5 | 50.72 | 45.27 | 32.85 |
| | +FastCAOTE | 11.65 | 21.2 | 37.92 | 24.47 | 20.32 | 13.25 | 32.11 | 21.75 | 24.84 | 67.5 | 78.07 | 40.27 | 5.17 | 25.21 | 50.17 | 46.37 | 32.52 |
| 8k | TOVA | 14.66 | 20.93 | 37.77 | 22.57 | 17.08 | 9.63 | 31.12 | 23.17 | 24.83 | 67 | 84.11 | 43.55 | 2.06 | 13.08 | 51.32 | 47.64 | 31.91 |
| | +CAOTE | 13.98 | 21 | 36.91 | 22.97 | 17.05 | 9.93 | 31.25 | 23.45 | 24.9 | 66.5 | 84.14 | 43.86 | 3.07 | 13.92 | 51.14 | 47.883 | 32.00 |
| | +FastCAOTE | 14.84 | 20.66 | 37.45 | 23.05 | 17.07 | 10.16 | 31.2 | 23.47 | 24.85 | 66.5 | 84.01 | 43.41 | 3.31 | 13.25 | 51.19 | 48.11 | 32.03 |
| | SnapKV | 12.76 | 20.88 | 37.1 | 22.49 | 18.19 | 13.83 | 31.33 | 23.37 | 24.8 | 65.5 | 84.88 | 44.49 | 5.2 | 35.83 | 51.31 | 47.82 | 33.74 |
| | +CAOTE | 13.66 | 20.41 | 38.08 | 24.76 | 17.31 | 13.21 | 31.3 | 23.62 | 24.82 | 66.5 | 84.88 | 44.16 | 5.17 | 36.58 | 51.24 | 47.94 | 33.98 |
| | +FastCAOTE | 14.56 | 20.99 | 37.61 | 25.56 | 18.02 | 13.89 | 31.37 | 23.27 | 24.83 | 66.5 | 84.88 | 44.01 | 5 | 35.92 | 51.13 | 48.14 | **34.11** |

Table 8: **Perplexity difference between different eviction methods with dense baseline.** Lower is better. Negative entry in table means the method performs better than dense baseline. The PPL of Qwen 2.5-3B-Instruct and Qwen 2.5-7B-Instruct is 8.4547 and 7.3188 respectively.

| Budget | H2O | | | TOVA | | | SnapKV | | |
| --- | --- | --- | --- | --- | --- | --- | --- | --- | --- |
| | | +CAOTE | +FastCAOTE | | +CAOTE | +FastCAOTE | | +CAOTE | +FastCAOTE |
| | | | | Qwen 2.5-7B-Instruct | | | | | |
| 2k | 0.4253 | 0.4422 | 0.3917 | 0.0567 | 0.059 | 0.1077 | 0.0369 | 0.0987 | **0.0307** |
| | | | | Qwen 2.5-3B-Instruct | | | | | |
| 2k | 0.2585 | 0.2168 | 0.2154 | 0.0603 | 0.0513 | 0.0507 | 0.0278 | 0.0199 | **0.0196** |

378 results (10) with budget of 6k and 8k. For Llama 3.1-8B-Instruct, *TOVA-FastCAOTE* performs
379 best for 6k budget, while *SnapKV-FastCAOTE* for 8k budget. For Llama 3.2-3B-Instruct *SnapKV-*
380 *FastCAOTE* performs best for both 6k and 8k budget. On the other hand, for Qwen 2.5-7B-Instruct,
381 *SnapKV-CAOTE* performs the best for both 6k and 8k, and for Qwen 2.5-3B-Instruct, *SnapKV-COATE*
382 performs best for 6k budget, and *SnapKV-FastCAOTE* performs best for 8k budget. Additionally, note
383 that all baseline token eviction methods achieve boost in accuracy when using *CAOTE* or *FastCAOTE*.

## C.2 Perplexity

385 We show perplexity results for Qwen 2.5 models in Table 8 for budget of 2k. *SnapKV-FastCAOTE*
386 performs best for both Qwen 2.5-3B-Instruct and 2.5-7B-Instruct, and using *CAOTE*, all methods
387 achieve improved perplexity.

## C.3 Needle in a Haystack

389 Table 9 shows Needle-in-haystack results for Qwen2.5 models for budget=6k. *SnapKV-FastCAOTE*
390 performs best for both Qwen 2.5-3B-Instruct and 2.5-7B-Instruct, and using *CAOTE*, all methods
391 achieve improved accuracy.

Table 9: **Needle-in-haystack accuracy** for Qwen 2.5-3B/2.5-7B-Instruct using baseline eviction methods with(out) *CAOTE*. Higher is better, maximum accuracy is 1.0.

| Budget | H2O | +CAOTE | +FastCAOTE | TOVA | +CAOTE | +FastCAOTE | SnapKV | +CAOTE | +FastCAOTE |
|---|---|---|---|---|---|---|---|---|---|
| | | | | | Qwen 2.5-7B-Instruct | | | | |
| 6k | 0.206 | 0.312 | 0.3 | 0.292 | 0.292 | 0.286 | 0.32 | 0.33 | **0.332** |
| | | | | | Qwen 2.5-3B-Instruct | | | | |
| 6k | 0.212 | 0.288 | 0.27 | 0.282 | 0.286 | 0.288 | 0.304 | 0.324 | **0.336** |

Table 10: **LongBench** accuracy for different tasks from each sub-category with varying block-prompt-size using Llama 3.2-3B-Instruct model with 4k budget. $\infty$ implies entire prompt is consumed without token eviction and eviction starts during generation. We faced out-of-memory issues with some inputs. * For $\infty$ case, HotPotQA gave out-of-memory on a 80GB A100 GPU and ran only on 114 out of 200 samples.

| Task | H2O | | | H2O-CAOTE | | | TOVA | | | TOVA-CAOTE | | | SnapKV | | | SnapKV-CAOTE | | |
|---|---|---|---|---|---|---|---|---|---|---|---|---|---|---|---|---|---|---|
| block-size | 32 | 128 | $\infty$ | 32 | 128 | $\infty$ | 32 | 128 | $\infty$ | 32 | 128 | $\infty$ | 32 | 128 | $\infty$ | 32 | 128 | $\infty$ |
| HotPotQA* | 20.43 | 24.49 | 35.71 | 32.37 | 40.28 | 46.37 | 44.36 | 45.55 | 52.93 | 44.72 | 45.25 | 52.94 | 45.99 | 46.7 | 53.67 | 44.66 | 45.64 | 52.67 |
| Qasper | 30.8 | 31.94 | 32.58 | 36.74 | 40.79 | 38.93 | 38.65 | 38.83 | 40.45 | 39.39 | 39.79 | 39.90 | 38.71 | 39.22 | 39.80 | 40.18 | 39.74 | 39.69 |
| GovReport | 5.52 | 5.44 | 7.07 | 30.10 | 30.82 | 31.48 | 29.32 | 29.41 | 31.25 | 29.24 | 29.39 | 31.15 | 29.62 | 29.79 | 31.82 | 30.15 | 30.07 | 32.42 |
| Trec | 46.50 | 53.00 | 56.00 | 59.00 | 65.5 | 72.00 | 62.00 | 64.00 | 71.50 | 64.00 | 63.00 | 71.50 | 64.00 | 65.50 | 71.50 | 67.50 | 67.50 | 71.50 |
| LCC | 51.59 | 51.23 | 51.51 | 54.06 | 55.32 | 54.51 | 56.23 | 55.64 | 56.34 | 56.18 | 55.68 | 56.39 | 56.24 | 55.74 | 56.56 | 56.27 | 55.73 | 56.59 |

## C.4 Ablations

**Difference in Attention Outputs** In Figure 3, we check the average change in attention score in different layers for various eviction methods to provide insight into the performance gain realized by *CAOTE*. We measure the difference between the dense attention-output and eviction method based attention output, both with and without *CAOTE*. We repeat this for each layer and average the result to verify that *CAOTE* results in smaller $L_2$ distance to dense attention-output. Figure 3 shows the normalized mean squared error between the dense attention output and eviction-based attention output for *H2O, TOVA* and *SnapKV*, both with and without *CAOTE* in dashed-lines and solid-lines respectively. We observe that the *CAOTE* variant always produces a smaller gap from dense attention output (no eviction) compared to baselines, supporting our claim that CAOTE evicts the token which minimizes deviation from true attention output. Furthermore, we show that discrepancy in attention output due to eviction (eviction error) causes discrepancy in logits in B.2; to emphasize that optimizing for eviction error results in less discrepancy in logits and, therefore, less deviation on downstream task accuracy.

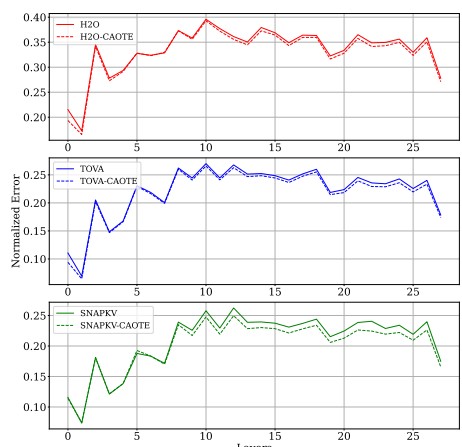

Figure 3: Normalized error between attention outputs with and without eviction for Llama 3.2-3B-Instruct, measured on 5 samples from Qasper dataset.

**Different Prompt Block Sizes** Prompt block size dictates how many tokens will be evicted at once. A moderately large prompt block size results in faster prompt processing but evicts more tokens each time, while a smaller prompt block size results in slower prefilling. To evaluate the impact of prompt block size, we evaluated a subset of LongBench tasks with block prompt sizes of 32, 128 and $\infty$, i.e., process the entire prompt as a single large block and display the results in Table 10 for Llama3.2 with budget=4k. Based on the CAOTE formulation (single token eviction), the smaller the block-size the better CAOTE would perform as seen for block-size=32 in Table 10, however, even for large prompt block size, CAOTE performs competitive to baseline if not better. Note that the infinite prompt block size initially violates the memory/budget constraint, and additionally, we observe out-of-memory errors for several samples for the HotPotQA and GovReport tasks when using infinite block size. This problem is not present with block-size=$\{32, 128\}$.

