# OpenReview forum: "CAOTE: Efficient Caching through Attention Output based Token Eviction"
_NeurIPS.cc/2025/Conference — Submitted to NeurIPS 2025_

### Official Review · Reviewer_CdVZ · 2025-06-20

**Clarity:** 2
**Significance:** 1
**Originality:** 2
**Rating:** 2
**Confidence:** 4

**Summary:**

The paper introduces CAOTE, a KV cache eviction method that based on the L2 difference of attention output which is different from previous methods that only considered attention scores. Additionally, the paper also proposes a faster version which directly use the average of the value vectors rather than the weighted version. Both CAOTE and its faster variant can be applied directly on top of existing KV Cache eviction method. The paper further shows the empirical evaluation results on Llama 3.1-8B-Instruct, Llama 3.2-3B-Instruct and, Qwen 2.5-3B-Instruct, Qwen 2.5-7B-Instruct over multiple language modeling tasks to show the effectiveness of their method.

**Questions:**

1. For each dataset, what is the average prompt length and generation length? It is possible that for some tasks a token budget from {2k, 4k, 6k, 8k} is basically full attention without this specification.
2. How do you deal with Group Query Attention which is used in the models you have evaluated with. Are you averaging the computed CAOTE score within the same KV group?
3. Are you doing KV cache eviction for every attention layer in the model?
4. How is your method different from [1]?
5. Even though the method you proposed can be easily applied on top of existing cache eviction method based on their customized score, this might deviate from your theoretical foundation derived in section 3.1 as the token scoring from H2O cannot be translated to the attention output deviation of the current decoding step. Do you have any motivation why such generalization is reasonable?

Reference:

1. Feng, Yuan, et al. "Identify Critical KV Cache in LLM Inference from an Output Perturbation Perspective." arXiv preprint arXiv:2502.03805 (2025).

**Ethical Concerns:**

["NO or VERY MINOR ethics concerns only"]

**Final Justification:**

The authors have addressed my questions on the prompt length and generation length on different datasets.

But the main concern here is that the eviction-based method typically has this non-negligible accuracy gap compared with the dense baseline, as in one example proposed by the author during rebuttal, the dense baseline is 32.25, whereas the best performing H2O+FastCAOTE is only 24.55. Accuracy drop is actually an important factor in a practical serving scenario, and this gap is non-trivial.

In addition, like on the Qasper dataset for Llama3.1-8B model (and other cases): Dense baseline: 47.00 TOVA: 44.55 TOVA+CAOTE: 43.88 TOVA+FastCAOTE: 44.32 CAOTE has slightly worse performance, which contradicts the claim of consistent improvements on baseline methods, but this is a minor point. The major concern is the accuracy gap between the proposed method and the dense baseline.

**Limitations:**

Yes

**Quality:**

2

**Strengths And Weaknesses:**

Strengths

- The sparse attention problem is important in the long context or long chain-of-thought reasoning setup.
- The empirical evaluations are comprehensive with multiple models and datasets.

Weaknesses:

- The evaluation section lacks lots of important details: see the question section.
- There is at least one prior work [1] as far as I am aware of that also focus on attention output deviation based KV cache eviction, which is not covered by this work.
- Pure KV cache eviction based methods still have a large performance gap compared to the dense baseline and the improvements provided by the proposed method is mostly marginal (or even worse as in Table 2,3) with respect to the baseline cache eviction algorithm like TOVA or SnapKV. And the retrieval accuracy is far from ideal for the relatively easier needle-in-the-haystack task. The accuracy drop might be more significant in more challenging tasks like RULER or reasoning tasks.

Reference:

1. Feng, Yuan, et al. "Identify Critical KV Cache in LLM Inference from an Output Perturbation Perspective." arXiv preprint arXiv:2502.03805 (2025).

---

> ### Author Rebuttal · Authors · 2025-07-30
>
> We thank the reviewer for their thoughtful feedback. Below, we address each of the reviewer’s points in detail:
>
> 1. **Average prompt and generation lengths per dataset**
>
> We will include these details in the appendix. Below are the average prompt and generation lengths across tasks:
>
> Prompt Lengths:
>
> - QA tasks: 4k – 16k
> - Summarization: 2.1k – 16k
> - Few-shot learning: 5.2k – 22.4k
> - Synthetic tasks: 7k – 11k
> - Code: 1.5k – 4.2k
> - Needle-in-the-haystack: up to 32k
>
> Generation Lengths:
> - QA tasks: 100–300 tokens
> - Summarization: 300–800 tokens
> - Multi-document reasoning: 500–1000 tokens
> - Code: ~500 tokens
>
> Thus, the total token usage is typically >4k, and often exceeds 6k or 8k, validating the relevance of our token budgets. As shown in the main paper, CAOTE consistently improves performance over baselines at 2k and 4k budgets.
>
> 2. **Handling Grouped Query Attention (GQA)**
>
> In our current implementation, we evict the same token index across heads. For GQA, we compute the maximum CAOTE score across groups for each token and use that for eviction. We will clarify this in the paper.
>
> Our method, like other token eviction strategies, can be extended to head-wise eviction, as explored in AdaKV.
>
> 3. **KV cache eviction per layer**
>
> Yes, we perform KV cache eviction at every attention layer, consistent with the baselines we compare against.
>
> 4. **Difference from [1] Feng et al. (2025)**
>
> Our method differs from [1] in both formulation and implementation:
>
> - We derive a closed-form expression for the perturbation in attention output due to token eviction, grounded in optimization theory.
> - In contrast, [1] proposes an upper bound on output perturbation, which is also discussed in [2].
> - Algorithm 1 in [1] uses scores of the form, $(A+\epsilon).|V|_1$, where $A$ is the attention scores and $V$ is the value vectors, which is a **heuristic approximation** involving value vectors. This is fundamentally different from our CAOTE score, which is derived from exact optimization for single-token eviction.
>
> References:
>
> [1] Feng, Yuan, et al. "Identify Critical KV Cache in LLM Inference from an Output Perturbation Perspective." arXiv:2502.03805 (2025)
>
> [2] Feng, Yuan, et al. "AdaKV: Optimizing KV Cache Eviction by Adaptive Budget Allocation for Efficient LLM Inference." arXiv:2407.11550 (2024)
>
> 5. **Generalizing CAOTE to other eviction methods**
>
> We agree that directly using CAOTE with methods like H2O or SnapKV may deviate from the theoretical assumptions in Section 3.1. However, we propose a reasonable generalization:
>
> - Many custom eviction scores (e.g., H2O, SnapKV) are functions of attention scores, and can be interpreted as effective attention scores.
> - We normalize these scores before applying CAOTE, which preserves token ranking but allows us to compute a meaningful CAOTE score using value vectors.
> - Simply using attention scores based on last token with CAOTE is same as TOVA + CAOTE, which already boosts performance, however, is not suitable for other eviction methods.
>
> Our empirical results show that this generalization is effective in practice, and we believe it opens up a flexible framework for combining CAOTE with a wide range of eviction strategies.
>
> 6. **On CAOTE’s performance and gap to dense baselines**
>
> We respectfully disagree with the claim that CAOTE leads to marginal or worse performance. In Tables 2 and 3, CAOTE consistently improves the performance of baseline methods.
>
> For example, on the Musique dataset with a 2k token budget:
>
> - Dense baseline: 32.25
> - H2O: 2.17
> - TOVA: 14.77
> - SnapKV: 16.53
> - H2O+CAOTE: 22.06
> - H2O+FastCAOTE: 24.55
>
> This shows a significant improvement over all baselines, including SoTA methods. We observe similar trends across other QA tasks like Qasper and MF-en.
>
> We believe CAOTE reduces the gap between dense inference and token-eviction-based methods, and we expect this to hold for more challenging tasks like RULER as well
>
> We thank the reviewer again for their thoughtful feedback and hope our clarifications and additional analyses address the concerns raised.

---

> > ### Author Response · Authors · 2025-08-03
> > **Follow-Up on Review Feedback (CdVZ)**
> >
> > I hope you're doing well!
> >
> > I just wanted to follow up on the responses I shared a couple of days ago regarding the review. I wanted to check if I’ve addressed all your questions or if there’s anything else I can clarify before the August 6 deadline.
> >
> > Please feel free to let me know if anything needs further input. I really appreciate your time and feedback!

---

> > ### Comment · Reviewer_CdVZ · 2025-08-05
> >
> > The main concern here is that the eviction-based method typically has this non-negligible accuracy gap compared with the dense baseline, as in your example, the dense baseline is 32.25, whereas the best performing H2O+FastCAOTE is only 24.55. Accuracy drop is actually an important factor in a practical serving scenario, and this gap is non-trivial.
> >
> > And in addition, like on the Qasper dataset for Llama3.1-8B model (and other cases):
> > Dense baseline: 47.00
> > TOVA: 44.55
> > TOVA+CAOTE: 43.88
> > TOVA+FastCAOTE:  44.32
> > CAOTE has slightly worse performance, which contradicts the claim of consistent improvements on baseline methods, but this is a minor point. The major concern is the accuracy gap between the proposed method and the dense baseline.
> >
> > I appreciate the authors' response. I will keep my score.

---

> > > ### Author Response · Authors · 2025-08-05
> > > **Response to Reviewer CdVZ**
> > >
> > > Thank you for your thoughtful and constructive feedback. We would like to clarify that the primary goal of our work is not to propose a new standalone token eviction method, but rather to introduce CAOTE as a plug-and-play meta-heuristic that can be applied to **enhance existing token eviction strategies**. Our focus is on improving the performance of sparse baseline methods like TOVA and H2O, rather than directly competing with dense baselines.
> > >
> > > We acknowledge the accuracy gap between dense and sparse methods, particularly under tight memory budgets such as 2k tokens. However, this gap is a known trade-off in practical serving scenarios. Our contribution lies in **reducing this gap** by improving the performance of existing sparse methods through CAOTE. For example, while the dense baseline achieves 32.25, the best-performing H2O+FastCAOTE reaches 24.55 — a significant improvement over the original H2O baseline.
> > >
> > > Importantly, this gap narrows substantially when the memory budget is increased. At a 4k token budget, **H2O+CAOTE achieves an accuracy of 30.89**, which is **very close to the dense baseline**, and significantly better than other sparse methods. This demonstrates CAOTE’s practical value in real-world deployments where memory budgets can be flexibly tuned.
> > >
> > > Regarding the Qasper dataset and the Llama3.1-8B model, we appreciate your observation. Please note that here as well **H2O+CAOTE: 46.08 which is the closest to dense baseline: 47.00.**
> > > While CAOTE may not always yield improvements in every configuration, it consistently enhances or maintains performance across a majority of settings. We will revise the manuscript to better reflect this nuance and avoid overgeneralizing the consistency of improvements.
> > >
> > > We hope this clarification helps position CAOTE as a **general-purpose enhancer** for token eviction methods, designed to be modular, adaptable, and effective across a range of baselines and budgets.

---

> > > > ### Comment · Reviewer_CdVZ · 2025-08-05
> > > >
> > > > I understand the proposed work is not trying to carry out a new sparse attention algorithm but rather trying to build on top of existing ones. But I don't think it makes sense to build on top of something that is not state-of-the-art in terms of the accuracy gap with dense baseline. I would recommend the authors to try apply this on some better performing methods like Quest [1]/DuoAttention [2]/TidalDecode [3] in a future revision.
> > > >
> > > > references:
> > > > 1. Tang, Jiaming, et al. "Quest: Query-aware sparsity for efficient long-context llm inference." arXiv preprint arXiv:2406.10774 (2024).
> > > >
> > > > 2. Xiao, Guangxuan, et al. "Duoattention: Efficient long-context llm inference with retrieval and streaming heads." arXiv preprint arXiv:2410.10819 (2024).
> > > >
> > > > 3. Yang, Lijie, et al. "Tidaldecode: Fast and accurate llm decoding with position persistent sparse attention." arXiv preprint arXiv:2410.05076 (2024).

---

> > > > > ### Author Response · Authors · 2025-08-05
> > > > > **Response to Reviewer CdVZ**
> > > > >
> > > > > Thank you again for your thoughtful feedback and for suggesting additional directions. We would like to respectfully clarify a few points:
> > > > >
> > > > > **Inclusion of State-of-the-Art Baselines**
> > > > >
> > > > > Contrary to the impression that our work does not include strong baselines, we do evaluate **SnapKV [4] +CAOTE**, which performs very close to the dense baseline across multiple datasets and budgets (see Tables 1, 2 with budget 4k and Tables 6, 7 with budget 6k). SnapKV is a recent and competitive method, and unlike Quest [1], it is evaluated extensively on **LLaMA3** models, whereas Quest uses **LLaMA2** and reports results on a limited subset of LongBench tasks. We therefore believe our choice of baselines is both relevant and representative of current state-of-the-art.
> > > > >
> > > > > **Orthogonality of DuoAttention and TidalDecode**
> > > > >
> > > > > Both DuoAttention [2] and TidalDecode [3] are **orthogonal to our focus**. These methods address attention head selection and using same token index across layers with cache correction strategies, respectively, whereas our work focuses on **token-level eviction**. While CAOTE could potentially be combined with these approaches, such integration is outside the scope of this paper. Our goal is to improve existing token eviction strategies in a modular and plug-and-play fashion.
> > > > >
> > > > > **Broader Model Coverage**
> > > > >
> > > > > Finally, we would like to highlight that our evaluation is **not limited to the LLaMA model family**. We also include results on **Qwen2.5**, demonstrating CAOTE’s generalizability across architectures. This broader coverage strengthens the practical relevance of our method.
> > > > >
> > > > > We appreciate your suggestions and will incorporate them into our future work section. Thank you again for helping us refine the scope and positioning of our contribution. We hope this clarifies your queries.
> > > > >
> > > > > Reference:
> > > > >
> > > > > [4] Li, Yuhong, et al. "Snapkv: Llm knows what you are looking for before generation." Advances in Neural Information Processing Systems 37 (2024): 22947-22970.

---

### Official Review · Reviewer_nboM · 2025-06-25

**Clarity:** 3
**Significance:** 3
**Originality:** 3
**Rating:** 3
**Confidence:** 4

**Summary:**

The paper proposes CAOTE, a post-training token-eviction strategy for LLM KV caches that augments traditional attention-score heuristics with information from value vectors. Existing methods treat the attention weight as a proxy for token importance, ignoring how each token’s value actually shapes the attention output, and are therefore sub-optimal.

CAOTE addresses this limitation by deriving a closed-form “eviction error’’—the expected L2 change in the attention output when a token is removed—and using it as a new importance score. The resulting CAOTE score fuses the attention weight with the value vector in a single analytic expression, yielding a simple yet principled criterion for cache eviction.

Because CAOTE is formulated as a meta-heuristic, it can be layered on top of any existing policy. Experimental results on LLama-3 and Qwen-2.5 across LongBench, perplexity evaluation and Needle-in-a-Haystack show that adding CAOTE systematically improves strong baselines such as H2O, TOVA and SnapKV, sometimes even reducing token count while raising accuracy.

**Questions:**

+ Your theoretical framework is built on single-token eviction. For the multi-token eviction used during block-wise prefill, could you elaborate on the sub-optimality of your greedy, independent-eviction assumption?
+ Could you provide a quantitative analysis of the computational overhead (e.g., extra FLOPs or percentage increase in latency per step) of both CAOTE and FastCAOTE?
+ As eviction decisions are made per layer, how does this error propagate and potentially compound in deeper models? Does your per-layer optimization effectively mitigate this compounding effect more than baselines?
+ How does CAOTE interact with adaptive budget allocation methods (e.g., AdaKV) that dynamically adjust cache size per layer?

**Ethical Concerns:**

["NO or VERY MINOR ethics concerns only"]

**Final Justification:**

After reading all the reviews and rebuttals, this paper can be further improved before it could be published at top-tier conference like NeurIPS, although the responses answers some of my concerns and there are indeed some good points I love.

**Quality:**

3

**Strengths And Weaknesses:**

Strengths
+ Novel contribution: CAOTE augments traditional, attention-only heuristics with value-vector information by explicitly minimizing the change in the attention output, yielding a conceptually clear and previously unexplored criterion for KV-cache eviction.
+ Solid theory: Theorem 3.2 formally shows that the CAOTE score equals the mean-squared error incurred in the attention output after evicting a token, giving the method a rigorous foundation.
+ Convincing empirical study: Extensive tests on LongBench, perplexity, and Needle-in-a-Haystack with Llama-3 and Qwen-2.5 demonstrate consistent improvements over strong baselines such as H2O, TOVA, and SnapKV.

Weaknesses
- Single-token assumption: The derivation assumes only one token is evicted, while the experiments use block-wise prefilling that discards multiple tokens at once. The theoretical consequences of joint eviction remain unanalysed.
- Overhead not quantified: The paper does not report the additional computation or latency introduced by computing CAOTE scores at inference time.
- FastCAOTE is heuristic: Replacing the attention output with the mean value vector is empirically effective but lacks a principled justification or error analysis.
- Limited model coverage: All experiments are on Llama-3 and Qwen-2.5; results on larger MoE models would strengthen the claim of generality.
- Incomplete related work: Recent methods such as CAKE (ICLR 2025) are not compared, leaving readers uncertain about CAOTE’s relative standing in the latest literature.

---

> ### Author Rebuttal · Authors · 2025-07-30
>
> We thank the reviewer for acknowledging the novelty and theoretical grounding of CAOTE. Below, we address each of the reviewer’s points in detail:
>
> 1. **On multi-token eviction and suboptimality during prefill**
>
> We agree that our theoretical framework is based on single-token eviction, and we acknowledge that block-wise prefill involves evicting multiple tokens simultaneously. In this case, our current implementation uses a greedy, independent-eviction strategy, which may not be globally optimal.
>
> We discuss the challenges associated with CAOTE's multi-token formulation. Below, we provide a detailed analysis of the sub-optimality of greedy, independent eviction and how CAOTE can be extended to handle multiple tokens jointly.
>
> To illustrate, consider evicting two tokens in order $i=1, j=2$. From CAOTE, the single-token eviction error for token 1 is:
> $$
> \frac{a_1}{1-a_1}||X_{attn}-v_1||_2
> $$
>
> After evicting token 1, the updated attention output becomes:
> $$
> X_{attn,[1]}=\Sigma_{i=2}^{n}a_{i}^{'}v_i, \text{where } a_{i}^{'}=\frac{a_i}{1-a_1}
> $$
>
> Now, evicting token 2 from this updated distribution yields:
> $$
> X_{attn,[1,2]}=\Sigma_{i=3}^{n}a_{i}^{''}v_i, \text{where } a_{i}^{''}=\frac{a_{i}^{'}}{1-a_{2}^{'}}=\frac{a_i}{1-a_1-a_2}
> $$
>
> This leads to the following insights:
> - The updated attention output can be expressed as:
> $$
> X_{attn,[1,2]}=\frac{1}{1-a_{2}^{'}}(X_{attn,[1]}-a_{2}^{'} v_2)
> $$
> - The eviction error for removing token 2 after token 1 is:
> $$
> ||X_{attn,[1]}-X_{attn,[1,2]}||_2
> $$
> - Substituting the expressions above, we obtain a **closed-form joint eviction score**:
> $$
> \frac{1}{1-a_1-a_2}||a_1(X_{attn}-v_1) + a_2(X_{attn}-v_2)||_2
> $$
>
> This formulation generalizes to evicting $m$ tokens jointly:
> $$
> \text{eviction error} = \frac{1}{1-\Sigma_{i=1}^{m}a_i}||\Sigma_{i=1}^{m}a_i(X_{attn}-v_i)||_{2}
> $$
>
> This expression highlights the combinatorial nature of multi-token eviction: for $n$ tokens and $m$-token eviction, there are $\binom{n}{m}$ possible combinations. This makes exact computation intractable for large $m$, motivating the need for approximate or greedy strategies.
>
> Importantly, existing token eviction methods typically rely only on attention scores and assume independence between tokens. As a result, the relative ranking of tokens remains unchanged even after evictions, which limits their effectiveness in multi-token settings.
>
> In contrast, CAOTE explicitly models the interplay between attention scores and value vectors, capturing how evicting one token affects the contribution of others. This introduces interdependencies between tokens during eviction, which are critical for accurate multi-token decisions.
>
> We are actively exploring connections to discrete optimization to develop efficient algorithms for joint multi-token eviction, potentially leveraging submodular optimization or greedy approximations with theoretical guarantees.
>
> 2. **Quantitative analysis of CAOTE and FastCAOTE overhead**
>
> We analyze the computational overhead of CAOTE and FastCAOTE during both prefill and generation phases. Let:
>
> - $s$: sequence length
> - $d$: hidden dimension
> - $d_{KV}$: KV hidden dimension
> - $d_{FFN}$: intermediate dimension
> - $d_V$: vocabulary size
> - $L$: number of hidden layers.
>
> Floating-point Operation count:
> - *prefill flops*: $2Lsd(d + d_{KV} + d_{FFN}+2s+6) + dd_V$
> - *generation flops*: $2Ld(d + d_{KV} + d_{FFN}+2s+6) + dd_V$
> - *CAOTE flops*: $Ls(7d+ 3)$
> - *FastCAOTE flops*: $Ls(4d+3)+L$
>
> Relative overhead for LLaMA3.1-8B:
>
> **Prefill Overhead Ratio**: (*CAOTE flops*)/(*prefill flops*)
> Sequence Length|ratio with CAOTE|ratio with FastCAOTE
> -|-----|---------
> 4k|1.6e-4|8.9e-5
> 8k|9.25e-5|5.28e-5
> 32k|6.45e-5|3.69e-5
>
> **Generation Overhead Ratio**: (*CAOTE flops*)/(*generation flops*)
> Sequence Length|ratio with CAOTE|ratio with FastCAOTE
> -|-----|---------
> 512|0.08|0.046
> 1024|0.15|0.087
>
> These results show that CAOTE adds negligible latency, especially during prefill, and FastCAOTE is even more efficient.
>
> 3. **On error propagation across layers**
>
> We refer the reviewer to Appendix B.2, where we relate the CAOTE score (eviction error) to downstream accuracy. Unlike other methods, CAOTE directly minimizes the change in attention output, which helps mitigate error propagation.
>
> Our theoretical analysis suggests that early-layer errors propagate more significantly, motivating per-layer optimization. In future work, we plan to explore joint optimization across layers and adaptive cache budgeting, where earlier layers may receive larger budgets to reduce cumulative error.
>
> 4. **Interaction with adaptive budget methods (e.g., AdaKV)**
>
> As discussed on lines 245–249, CAOTE complements AdaKV. While AdaKV dynamically adjusts cache size per layer based on attention scores, it does not consider value vectors. Integrating CAOTE into AdaKV can further improve eviction decisions by incorporating richer information.
>
> 5. **On empirical justification for FastCAOTE**
>
> FastCAOTE approximates the CAOTE score by replacing the attention output with the mean value vector. While heuristic, we provide **empirical evidence** for its effectiveness.
>
> We compare the value errors of both which is the main difference
> - CAOTE: $|X_{attn}-v_i|_2$
> - FastCAOTE: $|\mu(V)-v_i|_2$, where $\mu(V)$ is the mean value vector
>
> Using a sample from NarrativeQA, we compute Spearman correlations across all layers for LLama3.2-3B:
>
> `[0.99 0.98 0.91 0.97 0.92 0.88 0.85 0.82 0.81 0.9  0.84 0.92 0.83 0.86, 0.91 0.88 0.94 0.96 0.92 0.95 0.99 0.99 0.96 0.98 0.98 0.88 0.99 0.96]`
>
> These high correlations demonstrate that FastCAOTE closely tracks CAOTE, justifying its use as a lightweight approximation. We will include this analysis in the camera-ready version.
>
> 6. **On results with larger MoE models**
>
> We appreciate the reviewer’s suggestion. Our current focus has been on efficient frontier models such as LLaMA3 and Qwen2.5, which are well-suited for edge deployment scenarios where compute and memory constraints are critical.
>
> We agree that evaluating CAOTE on larger Mixture-of-Experts (MoE) models would further demonstrate its generality. However, many state-of-the-art MoE models (e.g., DeepSeekMoE, Mixtral) are significantly larger and less practical for edge use cases. That said, we plan to include results on at least one open-source MoE model in the camera-ready version to strengthen the empirical scope of our work.
>
> 7. **On comparison with CAKE (ICLR 2025)**
>
> Thank you for pointing this out. We will revise the Related Work section to include CAKE. Furthermore, CAOTE can be integrated with CAKE, which uses attention scores for slicing. Since CAOTE incorporates value vectors, the two methods are complementary and could be combined for further gains.
>
> We thank the reviewer again for their thoughtful feedback and hope our clarifications and additional analyses address the concerns raised.

---

> > ### Author Response · Authors · 2025-08-03
> > **Follow-Up on Review Feedback**
> >
> > I hope you're doing well!
> >
> > I just wanted to follow up on the responses I shared a couple of days ago regarding the review. I wanted to check if I’ve addressed all your questions or if there’s anything else I can clarify before the August 6 deadline.
> >
> > Please feel free to let me know if anything needs further input. I really appreciate your time and feedback!

---

> > > ### Comment · Reviewer_nboM · 2025-08-04
> > >
> > > I have no further questions.

---

> > > > ### Author Response · Authors · 2025-08-04
> > > > **Response to nboM**
> > > >
> > > > Thank you for your response and for taking the time to review our work. We appreciate your engagement and hope that our clarifications in the rebuttal were helpful.
> > > >
> > > > If our responses addressed your concerns, we kindly hope you were able to reflect that in your updated evaluation. Please let us know if there’s anything further we can clarify.

---

### Official Review · Reviewer_DSfh · 2025-07-05

**Clarity:** 2
**Significance:** 2
**Originality:** 3
**Rating:** 2
**Confidence:** 4

**Summary:**

This article primarily focuses on compressing the KV cache of long-context LLMs using a token-eviction-based approach (i.e., removing tokens from the context that have minimal impact on subsequent generation). The authors observe that many classic existing methods rely solely on attention scores to measure the importance of tokens. However, in the transformer’s attention layer, the output is a weighted sum of value states based on the attention scores. To address this, they propose a method that considers both attention scores and value states. Experimental results demonstrate the effectiveness of their approach.

**Questions:**

See the Weaknesses.

**Ethical Concerns:**

["NO or VERY MINOR ethics concerns only"]

**Limitations:**

Given the incomplete nature of the experiments, I believe there are still some limitations that have not been addressed. However, to some extent, this method is novel. In addition, the authors have not discussed the social impact of their work.

**Quality:**

2

**Strengths And Weaknesses:**

Strengths
- The KV cache compression is important since long-context is more and more necessary recently.
- The motivation is clear and significant. I agree with the authors’ view that value states should be taken into account when assessing the importance of tokens.

Weaknesses

- The writing is somewhat poor, especially in the Introduction and Method sections. Specifically, the Introduction devotes too much space to background and previous methods, with a serious lack of introduction to the authors’ own approach. For example, it merely states the motivation for using value states in addition to attention scores, but doesn’t explain at all how value states are actually utilized. In the Method section, Figure 1 only illustrates motivation, while the details of the method remain completely unclear.
- The experiments are rather lacking, as reflected in several aspects:
  - Baselines: As far as I know (though I may not be fully up-to-date on the latest work), there are many token-eviction-based methods that are not mentioned in this paper, including but not limited to StreamingLLM [1], Quest [2], CaM [3], and others. The baselines considered by the authors need to be more thoroughly investigated.
  - The scale of the experiments is also somewhat outdated. Current long-context LLMs are already capable of handling very long contexts (up to 1M tokens), so it is necessary to explore at least 128K tokens. Specifically, it is important to conduct experimental validation on InftyBench [4].
  - The depth of the experiments is insufficient. The authors do not analyze why CAOTE works—in other words, they do not investigate which specific changes compared to baseline methods (such as H2O) lead to performance improvements. The authors neither provide a qualitative analysis through case studies nor offer any quantitative experiments to support their claims. Furthermore, I observe that in Figure 3, CAOTE does not show any significant improvement over the baseline methods.
  - The authors do not discuss the impact of their method on inference latency.
  - There is a lack of adaptation to existing inference engines. Currently, there are many frameworks, such as VLLM and SGLANG, that can significantly improve LLM inference speed, but this method is not easily adaptable to them.

[1] Xiao, Guangxuan, et al. "Efficient streaming language models with attention sinks." arXiv preprint arXiv:2309.17453 (2023).

[2] Tang, Jiaming, et al. "Quest: Query-aware sparsity for efficient long-context llm inference." arXiv preprint arXiv:2406.10774 (2024).

[3] Zhang, Yuxin, et al. "Cam: Cache merging for memory-efficient llms inference." Forty-first International Conference on Machine Learning. 2024.

[4] Zhang, Xinrong, et al. "ınftyBench: Extending Long Context Evaluation Beyond 100K Tokens." ACL (1). 2024.

---

> ### Author Rebuttal · Authors · 2025-07-30
>
> We thank the reviewer for their thoughtful feedback and for recognizing the novelty of incorporating value vector information in token eviction. Below, we address each of the reviewer’s points in detail:
>
> 1. **Writing quality in Introduction and Method sections**
>
> We agree that the Introduction and Method sections can better highlight our contributions. In the camera-ready version, we will:
>
> - Condense background material to make space for a clearer and more focused introduction to CAOTE.
> - Add a concise explanation of how value vectors are integrated into the eviction score.
> - Expand Figure 1 to illustrate the full CAOTE pipeline, not just the motivation
>
> 2. **Incomplete experiments and social impact**
>
> We would like to clarify that our experimental setup aligns with recent token eviction papers. Importantly, CAOTE is not a competing method, but a boosting mechanism that improves the downstream accuracy of existing eviction strategies.
>
> Regarding social impact, we believe our work is highly relevant for edge deployment, where memory and compute constraints are critical. By improving KV-cache compression, CAOTE enables:
>
> - More accurate inference on edge devices,
> - Reduced energy consumption,
> - Enhanced user privacy by minimizing cloud dependency.
>
> We will include a dedicated discussion on social impact in the camera-ready version.
>
> 3. **Additional baselines: StreamingLLM, Quest, CAM**
>
> We appreciate the reviewer’s suggestions and clarify the following:
>
> - CAOTE is complementary, not competitive. It can be applied on top of existing methods to improve performance.
> - We included StreamingLLM (referred to as Sink in the paper) in Tables 6 and 7 (Appendix). It performs worse than TOVA, SnapKV, and their CAOTE-augmented versions.
> - H2O+Caote outperforms StreamingLLM on LLaMA3 and Qwen2.5-3B models on QA tasks for 2k budget: Musique, 2WikiMQA, Qasper, MF-en.
>
> We will:
>
> - Add StreamingLLM results for additional budgets (2k, 4k).
> - Include Quest and CaM in the Related Work section.
>
> **On Quest**: It uses approximate attention scores. We believe CAOTE can be combined with Quest by approximating value vectors (e.g., using min/max pooling) and computing approximate attention outputs to guide eviction.
>
> **On CaM**: It uses Eq. (14) based solely on attention scores. CAOTE can enhance this by incorporating value vector information. Note that CaM only reports results on LLaMA2, without comparisons to other eviction methods.
>
> We will explore and report Quest+CAOTE and CaM+CAOTE results in the camera-ready version.
>
> 4. **Scale of experiments**
>
> We agree that scaling to longer contexts is important. However, our focus is on edge deployment, where 1M-token contexts are currently impractical.
>
> We emphasize that:
>
> - Most baselines mentioned by the reviewer use outdated LLaMA2-7B models.
> - We use state-of-the-art models like LLaMA3 and Qwen2.5, which have shorter context lengths but are more relevant for current applications.
>
> Regarding InftyBench:
>
> - Many open-source models require fine-tuning (e.g., LongLoRA, Qwen-Long) to be compatible.
> - Once token eviction methods are evaluated on InftyBench, CAOTE can be applied to boost their performance.
>
> We plan to include InftyBench results using fine-tuned long-context models in the camera-ready version.
>
> 5. **On understanding why CAOTE works and clarification on Figure 3**
>
> (a) **Why CAOTE works**:
>
> We believe the primary reason why CAOTE boosts the performance of existing cache eviction methods is because it optimizes for the change in attention output and seamlessly combines attention scores with value vectors—an underexplored component in token eviction.
>
> Even though attention scores (or their variants) can be used to gauge the importance of a token, ultimately it is the value hidden state that is passed through the model layers and finally projected to logits to sample a token.
>
> CAOTE's optimization is grounded in strong theoretical backing, as shown in Theorem 3.1 in the main manuscript, which establishes the connection between eviction error and dependence on value vectors:
> $$
> \text{Eviction Error} \propto ||X_{attn} - v_i||_{2}
> $$
> This demonstrates how CAOTE computes the optimal eviction strategy.
>
> Additionally, we discuss the relationship between CAOTE and downstream task performance in Appendix B.2.
>
> Finally, the multitude of experiments on LongBench (QA, summarization, coding, etc.), perplexity, and retrieval tasks show that CAOTE consistently boosts the performance of existing token eviction methods.
>
> (b) **On Figure 3**:
> We acknowledge that in some cases, the performance gap between baseline methods and their CAOTE-augmented versions may appear small. This is expected when the baseline is already near a local optimum. However, CAOTE helps push toward a global optimum. The gains are more pronounced in challenging settings. For example, on the Musique dataset (context length ~15k), using LLaMA3.1-8B with a 2k token budget, we observe:
>
> - H2O: 8.78
> - TOVA: 14.77
> - SnapKV: 16.53
> - H2O+CAOTE: **22.06**
>
> This significant improvement supports our theoretical claims and demonstrates CAOTE’s practical value.
>
> 6. **On inference latency of CAOTE**
>
> We analyze the computational overhead of CAOTE and FastCAOTE during both prefill and generation phases. The overhead is minimal, especially for FastCAOTE.
>
> Let $s$ be sequence length, $d$ the hidden dimension, $d_{KV}$ the KV hidden dimension, $d_{FFN}$ the intermediate dimension, $d_V$ the vocabulary size, and $L$ the number of hidden layers.
>
> Floating-point Operation count:
> - *prefill flops*: $2Lsd(d + d_{KV} + d_{FFN}+2s+6) + dd_V$
> - *generation flops*: $2Ld(d + d_{KV} + d_{FFN}+2s+6) + dd_V$
> - *CAOTE flops*: $Ls(7d+ 3)$
> - *FastCAOTE flops*: $Ls(4d+3)+L$
>
> Relative overhead for LLaMA3.1-8B:
>
> **Prefill Overhead Ratio**: (*CAOTE flops*)/(*prefill flops*)
> Sequence Length|ratio with CAOTE|ratio with FastCAOTE
> -|-----|---------
> 4k|1.6e-4|8.9e-5
> 8k|9.25e-5|5.28e-5
> 32k|6.45e-5|3.69e-5
>
> **Generation Overhead Ratio**: (*CAOTE flops*)/(*generation flops*)
> Sequence Length|ratio with CAOTE|ratio with FastCAOTE
> -|-----|---------
> 512|0.08|0.046
> 1024|0.15|0.087
>
> These results show that CAOTE adds negligible latency, especially during prefill, and FastCAOTE is even more efficient.
>
> 7. **On adaptation with inference engines**
>
> We agree that adapting CAOTE to inference engines like vLLM and SGLANG is important. However, we note that these frameworks currently do not support token eviction methods such as H2O, TOVA, or SnapKV either.
>
> Once these methods are integrated into such engines, CAOTE can be seamlessly adopted as it builds on top of them. While this is outside the scope of the current paper, we view it as a valuable direction for future work.
>
> We thank the reviewer again for their thoughtful feedback and hope our clarifications and additional analyses address the concerns raised.

---

> > ### Author Response · Authors · 2025-08-03
> > **Follow-Up on Review Feedback**
> >
> > I hope you're doing well!
> >
> > I just wanted to follow up on the responses I shared a couple of days ago regarding the review. I wanted to check if I’ve addressed all your questions or if there’s anything else I can clarify before the August 6 deadline.
> >
> > Please feel free to let me know if anything needs further input. I really appreciate your time and feedback!

---

> > > ### Author Response · Authors · 2025-08-06
> > > **Follow-up on Review Comments and Score**
> > >
> > > I hope you're doing well. I wanted to follow up on our previous response to your review comments. We believe we’ve addressed all the points you raised and would really appreciate your feedback when you have a chance.
> > >
> > > If everything looks good from your side, we’d also be grateful if you could consider updating your score to reflect the revisions, if appropriate.
> > >
> > > Please let us know if there’s anything further we can clarify or improve. We truly appreciate your time and support.

---

> ### Comment · Area_Chair_7UNY · 2025-08-07
>
> Dear reviewer DSfh,
>
> This is a kind reminder that reviewers should participate in discussions with authors. Please provide explanations about your updated (or not-updated) score and submit the “Mandatory Acknowledgement”. Please try to engage with the authors based on their rebuttal before the deadline. Your feedback is essential for the decision on this paper.
>
> Thank you!
>
> AC

---

### Official Review · Reviewer_V5hN · 2025-07-14

**Clarity:** 2
**Significance:** 3
**Originality:** 3
**Rating:** 3
**Confidence:** 3

**Summary:**

The paper present CAOTE (KV Caching through Attention Output Error-based Token Eviction), a post-training token eviction method for reducing memory usage in long-context LLM inference. Unlike existing methods that rely solely on attention scores to determine token importance, CAOTE directly minimizes the change in attention output (eviction error) caused by removing a token. This error can be computed in closed-form using both attention scores and value vectors. CAOTE can also be combined with existing score-based eviction strategies (e.g., H2O, TOVA, SnapKV) via a normalization trick, and an efficient approximation called FastCAOTE is proposed. Extensive experiments across LongBench, BookSum, and Needle-in-a-Haystack show that CAOTE consistently improves accuracy and perplexity, with up to 30–60% higher retrieval accuracy at the same memory budget

**Questions:**

- Could you provide statistical significance or variability measures to validate that CAOTE’s performance gains
- how would CAOTE extend to multi-token eviction scenarios

**Ethical Concerns:**

["NO or VERY MINOR ethics concerns only"]

**Limitations:**

no negative societal impact

**Quality:**

3

**Strengths And Weaknesses:**

stength:
- The method is mathematically solid becasue it defines eviction error as the L2 distance between full and evicted attention outputs and shows this equals a closed-form score computable from attention weights and values.
- enables more effective KV-cache trimming under tight memory constraints. For example, H2O+CAOTE boosts accuracy by over 30% on LongBench compared to H2O alone at 2k budget

weakness:
- CAOTE assumes single-token eviction at each step; during prefilling (multi-token insertions), the independence assumption may reduce optimality
- Some notation is dense and can benefit from simplification or intuitive explanation, the role of each term in Eq. (8) could be explained better

---

> ### Author Rebuttal · Authors · 2025-07-30
>
> We thank the reviewer for highlighting the strengths of our work. Below, we provide detailed responses to each of the reviewer’s questions:
>
> 1. **Statistical significance and variability of CAOTE’s performance**
>
> We appreciate the suggestion to include variability measures. In the paper, H2O+CAOTE consistently shows strong performance, often surpassing TOVA and SnapKV. We refer the reviewer to Tables 2 and 3, particularly under the 2k token budget for QA tasks such as Qasper, MF-en, 2WikiMQA, and Musique, where H2O+CAOTE achieves the best results.
>
> To further support this, we analyzed all baseline eviction methods with (and without) CAOTE on perplexity. We computed the difference between accuracy (or perplexity) per sample of a baseline method against baseline+CAOTE (and baseline+FastCAOTE). We report the mean, standard-deviation, minimum and maximum of this **difference**. We observe that with CAOTE the mean is always positive (showing gains), additionally the maximum gain ($\Delta$ Max Positive) is always much greater than the maximum loss ($\Delta$ Max Negative).
>
> **Statistics on perplexity improvement (decrease) for Lama3.2-3B with 2k budget**
> Method|$\Delta$Mean|$\Delta$Std|$\Delta$Max Negative|$\Delta$Max Positive|\% samples improved|
> ------|----|---|---|---|------------|
> H2O+CAOTE|0.2527|0.6394|1.0368|3.3395|68
> H2O+FastCAOTE|0.2502|0.6532|1.2082|3.0028|68
> ||
> TOVA+CAOTE|0.0510|0.1309|0.3068|0.6194|58
> TOVA+FastCAOTE|0.0609|0.1356|0.3162|0.6263|65
> ||
> SnapKV+CAOTE|0.1040|0.219|0.292|1.185|70
> SnapKV+FastCAOTE|0.1208|0.2322|0.2574|0.9162|71
>
> Therefore, sample-wise analysis shows that CAOTE (and FastCAOTE) help decrease downstream perplexity.
>
> We will include additional variability analyses in the camera-ready version.
>
>
> 2. **Extension to multi-token eviction**
>
> Extending CAOTE to multi-token eviction is an exciting direction, and we appreciate the reviewer’s interest in this aspect. Below, we provide a detailed analysis of the sub-optimality of greedy, independent eviction and how CAOTE can be extended to handle multiple tokens jointly.
>
> To illustrate, consider evicting two tokens in order $i=1, j=2$. From CAOTE, the single-token eviction error for token 1 is:
> $$
> \frac{a_1}{1-a_1}||X_{attn}-v_1||_2
> $$
>
> After evicting token 1, the updated attention output becomes:
> $$
> X_{attn,[1]}=\Sigma_{i=2}^{n}a_{i}^{'}v_i, \text{where } a_{i}^{'}=\frac{a_i}{1-a_1}
> $$
>
> Now, evicting token 2 from this updated distribution yields:
> $$
> X_{attn,[1,2]}=\Sigma_{i=3}^{n}a_{i}^{''}v_i, \text{where } a_{i}^{''}=\frac{a_{i}^{'}}{1-a_{2}^{'}}=\frac{a_i}{1-a_1-a_2}
> $$
>
> This leads to the following insights:
> - The updated attention output can be expressed as:
> $$
> X_{attn,[1,2]}=\frac{1}{1-a_{2}^{'}}(X_{attn,[1]}-a_{2}^{'} v_2)
> $$
> - The eviction error for removing token 2 after token 1 is:
> $$
> ||X_{attn,[1]}-X_{attn,[1,2]}||_2
> $$
> - Substituting the expressions above, we obtain a **closed-form joint eviction score**:
> $$
> \frac{1}{1-a_1-a_2}||a_1(X_{attn}-v_1) + a_2(X_{attn}-v_2)||_2
> $$
>
> This formulation generalizes to evicting $m$ tokens jointly:
> $$
> \text{eviction error} = \frac{1}{1-\Sigma_{i=1}^{m}a_i}||\Sigma_{i=1}^{m}a_i(X_{attn}-v_i)||_{2}
> $$
>
> This expression highlights the combinatorial nature of multi-token eviction: for $n$ tokens and $m$-token eviction, there are $\binom{n}{m}$ possible combinations. This makes exact computation intractable for large $m$, motivating the need for approximate or greedy strategies.
>
> Importantly, existing token eviction methods typically rely only on attention scores and assume independence between tokens. As a result, the relative ranking of tokens remains unchanged even after evictions, which limits their effectiveness in multi-token settings.
>
> In contrast, CAOTE explicitly models the interplay between attention scores and value vectors, capturing how evicting one token affects the contribution of others. This introduces interdependencies between tokens during eviction, which are critical for accurate multi-token decisions.
>
> We are actively exploring connections to discrete optimization to develop efficient algorithms for joint multi-token eviction, potentially leveraging submodular optimization or greedy approximations with theoretical guarantees.
>
> 3. **Clarification of notation in Eq. (8)**
>
> Thank you for the feedback. We will revise the notation in Eq. (8) and elsewhere to improve clarity. Specifically, all terms used in Eq. (8) are defined in the paragraph immediately preceding it (lines 124–126). In the camera-ready version, we will add more intuitive explanations to aid understanding
>
>
> We thank the reviewer again for their thoughtful feedback and hope our clarifications and additional analyses address the concerns raised.

---

> > ### Author Response · Authors · 2025-08-03
> > **Follow-Up on Review Feedback (V5hN)**
> >
> > I hope you're doing well!
> >
> > I just wanted to follow up on the responses I shared a couple of days ago regarding the review. I wanted to check if I’ve addressed all your questions or if there’s anything else I can clarify before the August 6 deadline.
> >
> > Please feel free to let me know if anything needs further input. I really appreciate your time and feedback!

---

> > > ### Author Response · Authors · 2025-08-06
> > > **Follow-up on Review Comments and Score**
> > >
> > > I hope you're doing well. I wanted to follow up on our previous response to your review comments. We believe we’ve addressed all the points you raised and would really appreciate your feedback when you have a chance.
> > >
> > > If everything looks good from your side, we’d also be grateful if you could consider updating your score to reflect the revisions, if appropriate.
> > >
> > > Please let us know if there’s anything further we can clarify or improve. We truly appreciate your time and support.

---

> ### Comment · Area_Chair_7UNY · 2025-08-07
>
> Dear reviewer V5hN,
>
> This is a kind reminder that reviewers should participate in discussions with authors. Please provide explanations about your updated (or not-updated) score and submit the “Mandatory Acknowledgement”. Please try to engage with the authors based on their rebuttal before the deadline. Your feedback is essential for the decision on this paper.
>
> Thank you!
> AC

---

> > ### Author Response · Authors · 2025-08-08
> > **Follow-up on Review Comments and Score**
> >
> > Dear reviewer V5hN
> >
> > I hope you're doing well. I wanted to follow up on our previous response to your review comments. We believe we’ve addressed all the points you raised and would really appreciate your feedback when you have a chance.
> >
> > If everything looks good from your side, we’d also be grateful if you could consider updating your score to reflect the revisions, if appropriate.
> >
> > Please let us know if there’s anything further we can clarify or improve. We truly appreciate your time and support.

---

### Comment · Area_Chair_7UNY · 2025-08-06

Dear Reviewer,

Please review the authors’ rebuttal, engage in discussions, and finalize your scores. Please write explanations about your updated (or not-updated) scores and submit the Mandatory Acknowledgement. If you have done so, thank you!

Your effort is greatly appreciated for the conference.

Thanks, AC

---

### Note · Authors · 2025-08-12

We sincerely thank the reviewers and Area Chair for their feedback and engagement with our submission. We have made every effort to address all reviewer comments thoroughly and constructively.

Reviewer V5hN: We addressed concerns by providing a detailed statistical analysis of CAOTE’s performance gains, including sample-wise variability metrics across multiple baselines. We elaborated on CAOTE's extension to multi-token eviction scenarios and clarified notation in Eq. (8)

Reviewer DSfh: We revised the Introduction and Method sections to better highlight CAOTE’s contributions and clarified its integration of value vectors. We expanded experiments to include StreamingLLM (Table 6,7) and plan to add methods such as Quest, CaM in Related Work. While our focus is on edge deployment, we will include InftyBench results using fine-tuned models. We also discuss CAOTE’s theoretical foundation, practical impact, and social relevance

Reviewer nboM: We addressed the single-token assumption by deriving a closed-form joint eviction score for multi-token settings. We quantified the compute overhead of (Fast)CAOTE, showing that both introduce minimal latency, especially during prefill. We discussed how CAOTE’s per-layer optimization mitigates error propagation and complements adaptive budget methods like AdaKV. We also justified FastCAOTE emperically via high Spearman correlations with CAOTE scores. While focused  on efficient frontier models, we plan to include results on open-source MoE models and comparisons to recent methods like CAKE.

Reviewer cdVZ: We clarified prompt and generation lengths across tasks, validating our token budgets. We addressed GQA and confirmed that eviction is performed per attention layer. We distinguished CAOTE from prior work [Feng et al., 2025] by emphasizing our closed-form derivation and optimization-based formulation. We explained our normalization strategy when combining CAOTE with other eviction methods and provided empirical evidence of its effectiveness, showing that for QA tasks in LongBench, H2O+CAOTE outperforms recent SoTA like SnapKV

Taken together, our responses demonstrate that CAOTE is a theoretically grounded, practically effective, and broadly applicable method for KV cache eviction. It consistently improves performance across models and tasks, introduces minimal overhead, and opens new directions for efficient long-context inference. We believe these contributions make CAOTE a valuable addition to the NeurIPS community.

---

### Decision · Program_Chairs · 2025-09-17

**Decision:**

Reject

**Comment:**

The work considered efficient token eviction for LLMs. It proposed a method, CAOTE, which optimizes the error due to removing a token. The error can be computed in closed form and thus leads to a criterion using attention scores and value vectors. The method can be used as a framework where existing attention score-based methods can be plugged in. Experiments show that the effectiveness of the method in improving the accuracy at the same memory budget.

While the topic is interesting and important, there are some common concerns.
1. Single-token assumption. The derivation is based on evicting only one token. The authors have shown the challenges in multi-token eviction, but haven't fully addressed the reviewers' concern (eg, whether the greedy approach yields a good approximation).
2. Evaluation scope. The reviewers would like more thorough evaluations (more baselines, models, and datasets, and ablations, etc). The authors have clarified what baselines/models/datasets have been used, and argued they support the conclusions. However, reviewers in general believe that more experimental results are needed for the publication.

Overall, the work is promising, while more thorough analysis and experiments can significantly strengthen the work.